# Mesopontine cholinergic inputs to midbrain dopamine neurons drive stress-induced depressive-like behaviors

Sebastian P. Fernandez[1,2], Loïc Broussot[1,2], Fabio Marti [3,4], Thomas Contesse[1,2], Xavier Mouska[1,2], Mariano Soiza-Reilly [3,5], Hélène Marie[1,2], Philippe Faure [3,4] & Jacques Barik[1,2]

Stressful life events are primary environmental factors that markedly contribute to depression by triggering brain cellular maladaptations. Dysregulation of ventral tegmental area (VTA) dopamine neurons has been causally linked to the appearance of social withdrawal and anhedonia, two classical manifestations of depression. However, the relevant inputs that shape these dopamine signals remain largely unknown. We demonstrate that chronic social defeat (CSD) stress, a preclinical paradigm of depression, causes marked hyperactivity of laterodorsal tegmentum (LDTg) excitatory neurons that project to the VTA. Selective chemogenetic-mediated inhibition of cholinergic LDTg neurons prevent CSD-induced VTA DA neurons dysregulation and depressive-like behaviors. Pro-depressant outcomes are replicated by pairing activation of LDTg cholinergic terminals in the VTA with a moderate stress. Prevention of CSD outcomes are recapitulated by blocking corticotropin-releasing factor receptor 1 within the LDTg. These data uncover a neuro-circuitry of depressive-like disorders and demonstrate that stress, via a neuroendocrine signal, profoundly dysregulates the LDTg.

[1] Université Côte d'Azur, Nice 06560, France. [2] Institut de Pharmacologie Moléculaire & Cellulaire, CNRS, UMR7275 Valbonne, France. [3] Université Pierre et Marie Curie, Paris 75005, France. [4] Neurosciences Paris Seine, INSERM U1130, CNRS, UMR 8246 Paris, France. [5] Institut du Fer à Moulin, INSERM, UMRS-839 Paris, France. Correspondence and requests for materials should be addressed to S.P.F. (email: fernandez@ipmc.cnrs.fr) or to J.B. (email: barik@ipmc.cnrs.fr)

D epression is a frequently diagnosed mental condition that has detrimental impact on psychological well-being and on health systems. It is the leading cause of disability worldwide and the World Health Organization estimates that 350 millions of people of all ages suffer from this disorder[1]. Social withdrawal and anhedonia, i.e., inability to experience pleasure from normally rewarding actions, are conspicuous manifestations of major depression[2]. These symptomatic facets of the disease have been associated with a dysregulation of the brain reward system, including dopamine (DA) neurons in the ventral tegmental area (VTA)[3,4].

This system has a predominant role in processing salient stimuli, both rewarding and aversive, to guide a broad range of adaptive behaviors[5]. DA neurons undergo experience-dependent plasticity, which is thought to be a cardinal cellular mechanism shaping behavioral strategies[6]. The use of preclinical models spanning from rodents to non-human primates is essential to apprehend the neurobiological complexity of depression[7]. For example, mice subjected to repeated bouts of social subordination exhibit a marked and sustained increase in the firing activity of VTA DA neurons[8–11]. Defeated mice display a wide range of depressive-like symptoms, including social aversion and anhedonia, which can be reversed by restoring normal VTA function[8,9]. These observations set the basis for a neuroanatomical specificity causally linking DA cellular adaptations and maladaptive behaviors to chronic stress. However, a clear understanding of how cellular adaptations within delineated neural circuits give rise to depressive-like behaviors is still largely lacking. In particular, identifying relevant inputs that shape DA neurons' responsiveness to stress is essential and has important implications for future therapeutic advances.

Among the many inputs received by VTA neurons[12], excitatory cholinergic and glutamatergic projections from the laterodorsal tegmentum (LDTg) are key to shape DA neurons' activity in response to salient stimuli[13,14]. Pharmacological inhibition or excitotoxic lesions of the LDTg are detrimental to VTA DA neurons' activity[13,15]. LDTg inputs to the VTA are key to shape reward responses[14,16], but their implication in stress-related experiences has yet to be addressed. We hypothesized that LDTg excitatory drive to VTA DA neurons could actively orchestrate cellular and behavioral manifestations of depressive-like behaviors. To test this, we combined retrograde labeling, and electrophysiological and behavioral approaches to assess the impact of a chemogenetic-based remote control of LDTg neurons' activity in mice subjected to a chronic social defeat (CSD) stress[17,18]. Our data indicate that CSD, via increase of corticotropin-releasing factor (CRF), triggers a hyperactivity of LDTg cholinergic neurons projecting to the VTA. This increased cholinergic tone is a prerequisite for the subsequent stress-induced cellular adaptations of VTA DA neurons and the appearance of social aversion and anhedonia.

## Results

**Silencing LDTg prevents maladaptations to chronic stress.** A single social defeat session exerted strong activation of excitatory LDTg neurons as evidenced by an increase in c-Fos expression, a classical marker of neuronal activity. Indeed, more than 5-fold c-Fos increase was detected in both cholinergic and glutamatergic LDTg neuronal populations (Fig. 1a). To interrogate whether the LDTg is embedded in the brain circuits that underlie the appearance of depressive-like disorders, we used a chemogenetic approach to remotely inhibit the LDTg before each daily defeat session of the chronic defeat paradigm. Wild-type (WT) mice were injected with AAV-hsyn-hM4-mcherry into the LDTg (hereafter called LDTg$^{hM4}$ mice) and allowed a 3-week recovery

period to achieve sufficient expression reaching 71% of LDTg neurons (Supplementary Fig. 1). This approach allowed silencing of transduced LDTg neurons in the presence of the agonist clozapine N-oxide (CNO) (Supplementary Fig. 2a). Acute chemogenetic-mediated inhibition of the LDTg via CNO injection, significantly decreased spontaneous discharge frequency (~40%) and bursting activity (~80%) in all VTA DA neurons juxtacellularly recorded in vivo in anesthetized mice (Fig. 1b). Of note, acute injection of CNO did not alter in vivo firing and bursting activity of VTA DA neurons (Supplementary Fig. 3a, b) in mice injected with AAV-hSyn-GFP into the LDTg (hereafter called LDTg$^{GFP}$ mice). This first set of data shows that the LDTg is activated by stress, and that its remote chemogenetic inhibition has a functional impact on VTA DA neurons activity. Next, LDTg$^{hM4}$ mice were submitted to 10 days of social defeat and received an intraperitoneal (i.p.) injection of saline or CNO (1 mg/kg) 30 min before each defeat session. Naive mice were treated accordingly but without defeat (Fig. 1c). All animals were tested in a drug-free condition. As expected, defeated mice treated with saline showed strong social avoidance (Fig. 1d) and anhedonia, as evidenced by a lack of sucrose preference compared with non-stressed saline-treated mice (Fig. 1e). In contrast, LDTg inhibition during CSD prevented stress impact, as defeated CNO-treated mice showed social interaction and sucrose preference that were comparable to that of undefeated animals (Fig. 1d, e). These effects cannot be attributed to a direct action of CNO, as LDTg$^{GFP}$ mice treated accordingly showed the expected stress-induced social aversion and anhedonia (Supplementary Fig. 3c). As these stress-related symptoms have been causally linked to pathophysiological cellular maladaptations in the DA system[8,9], we assessed VTA DA neuron firing and plasticity in both conditions using whole-cell current-clamp recordings. In acute brain slices from defeated saline-treated mice, VTA DA neurons exhibited increased excitability to current injection and higher AMPA-R/NMDA-R ratio when compared with undefeated mice (Fig. 1f, g). Mirroring the behavioral outcomes of LDTg inhibition, CSD failed to elicit VTA cellular adaptations. Indeed, no significant differences were observed in excitability or synaptic plasticity in VTA DA neurons from defeated LDTg$^{hM4}$ mice treated with CNO during CSD compared with undefeated mice (Fig. 1f, g, respectively). Of note, chronic CNO injections in naive and defeated LDTg$^{GFP}$ mice did not alter excitability of VTA DA neurons (Supplementary Fig. 3d). Altogether, these results indicate that LDTg is activated during chronic stress, and that LDTg inhibition prevents the development of depressive-like behaviors and the underlying cellular dysregulations of the VTA DA system.

**Chronic stress drives hyperexcitability of LDTg neurons.** The LDTg is heterogeneous containing non-overlapping VTA-projecting neurons[14] and converging evidence suggest that both cholinergic and glutamatergic LDTg neurons provide excitatory inputs to VTA DA neurons[12,16,19]. Thus, we hypothesized that these excitatory projections could be affected by chronic stress consequently leading to hyperactivity of VTA DA neurons. To unambiguously perform whole-cell patch-clamp recordings from cholinergic LDTg neurons that project to the VTA (LDTg$^{\rightarrow VTA}$), we injected green retrobeads in the VTA of ChAT-Cre × tdTomato mice (ChAT$^{tdTom}$ mice Fig. 2a), in which cholinergic neurons are tagged with tdTomato. To record from glutamatergic LDTg$^{\rightarrow VTA}$ neurons, we injected green retrobeads in the VTA and AAV-hsyn-DIO-mcherry in the LDTg of Vglut2-Cre mice[20] (Fig. 2b). Cholinergic and glutamatergic LDTg$^{\rightarrow VTA}$ neurons showed different excitability profile to depolarizing current steps, as well as different soma sizes, passive membrane properties, and action potential shape (Fig. 2c and Supplementary Fig. 4). This is

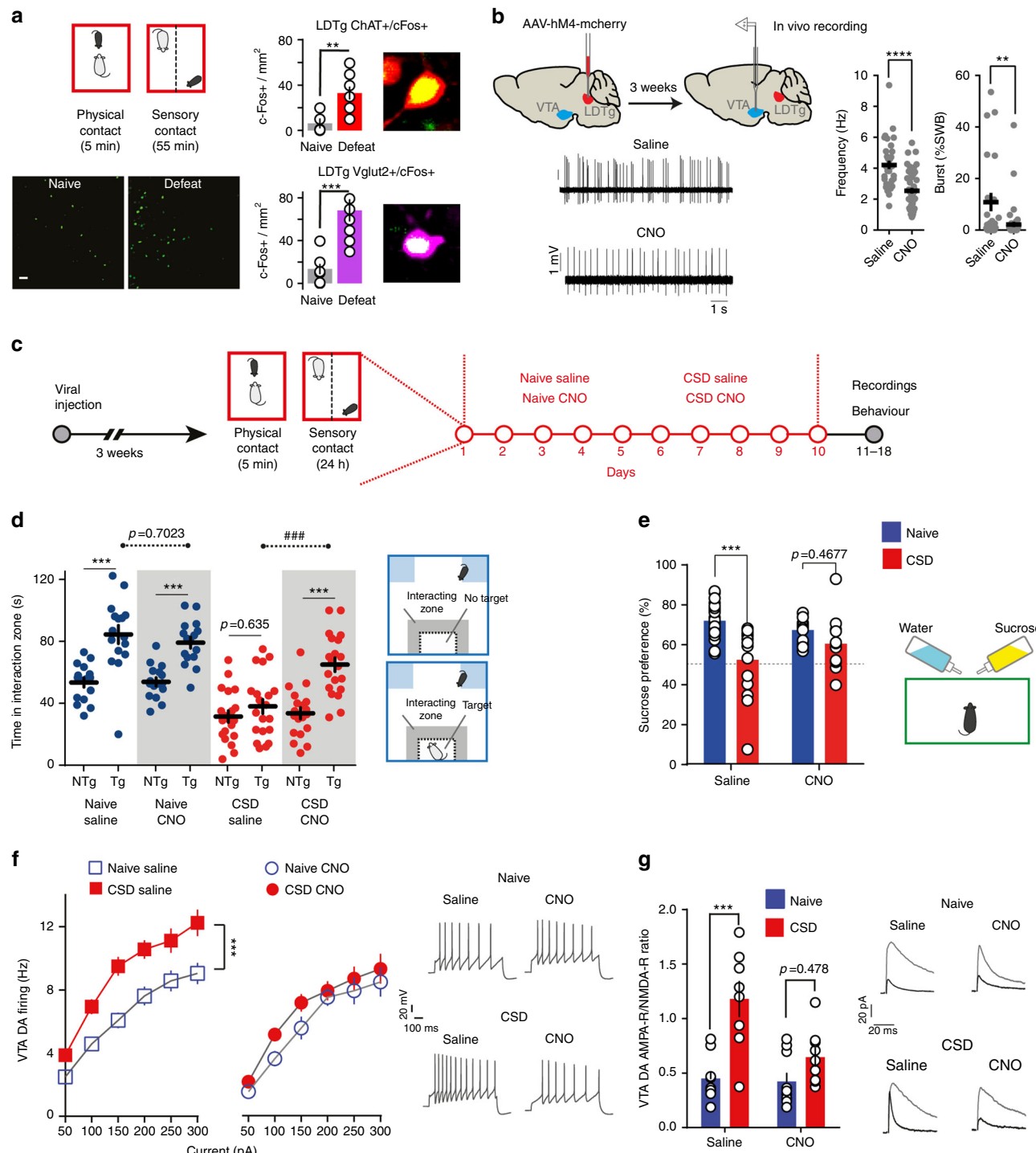

consistent with two non-overlapping neuronal populations that provide separate excitatory drive to midbrain DA neurons. CSD triggered a robust increase in excitability in both cholinergic and glutamatergic LDTg→VTA neurons. Indeed, cholinergic LDTg→VTA neurons responded with higher discharge frequency to depolarizing current injections in slices from defeated mice compared with undefeated animals (Fig. 2d, e). Similarly, glutamatergic LDTg→VTA neurons showed higher intrinsic excitability after CSD than undefeated mice (Fig. 2d–f). These results suggest that CSD increases excitatory drive from LDTg cholinergic and glutamatergic neurons that innervate the VTA, possibly driving the cellular hyperexcitability of midbrain DA cells.

**LDTg cholinergic neurons silencing dampens stress impact**. To ascertain the individual contribution of LDTg cholinergic and/or glutamatergic neurons to depressive-like behaviors and VTA DA dysregulations, we used a Cre-dependent chemogenetic approach to specifically silence these neuronal populations. First, AAV-hSyn-DIO-hM4-mcherry was stereotaxically injected in the LDTg of ChAT-Cre mice (hereafter called LDTg[ChAT-hM4] mice, Fig. 3a and Supplementary Fig. 5a). Selective inhibition of LDTg cholinergic neurons during CSD prevented the behavioral manifestations of stress. Indeed, although defeated LDTg[ChAT-hM4] mice treated with saline showed strong social avoidance and anhedonic responses to sucrose solution, CNO-treated LDTg[ChAT-hM4] mice

**Fig. 1** Chemogenetic inhibition of the LDTg during chronic defeat prevents depressive-like behaviors and VTA DA cellular adaptations. **a** Activation of LDTg cholinergic and glutamatergic neurons after a single defeat. Images shows LDTg c-Fos positive (c-Fos + ) cells (scale bar = 50 μm). Quantification of LDTg c-Fos + cells after acute defeat stress ($n = 3$ slices/mouse, 4 mice/condition, **$P < 0.001$ unpaired $t$-test). **b** In vivo recordings were performed in the VTA 3 weeks after viral injection. While under anesthesia, mice were given intravenously saline or CNO and changes in activity from baseline assessed. In all instances, dopamine neurons in the VTA showed a decrease in firing frequency (****$P < 0.0001$, $t$-test) and bursting activity (**$P < 0.01$, Mann–Whitney) in response to CNO when compared with saline (number of cell/mice: saline $n = 33/4$ and CNO $n = 45/5$). Voltage traces are shown. **c** Schematic experimental time line. **d** Social interaction times in the absence no target (NTg) or presence target (Tg) of an unfamiliar mouse. Defeated LDTg[hM4] mice treated with saline show marked social avoidance, which is not present in animals that received CNO before each defeat session (number of mice: Naive/Sal = 16; Naive/CNO = 16; CSD/Sal = 20; CSD/CNO = 19). Interaction treatment × target $F(3,67) = 5.14$, $P = 0.003$, Interaction treatment × stress $F(1,67) = 13.55$, $P < 0.001$; repeated-measures two-way ANOVA followed by Sidak's comparisons test, asterisks depict NTg vs. Tg comparisons, hash depicts comparisons between saline vs. CNO in the Tg condition; ***$P < 0.001$, ###$P < 0.001$. **e** Sucrose consumption is decreased in defeated saline-treated LDTg[hM4] mice but not in CNO-treated mice (number of mice: Naive/Sal = 23; Naive/CNO = 18; CSD/Sal = 16; CSD/CNO = 15). Interaction treatment × stress $F(1,66) = 5.39$, $P = 0.02$; two-way ANOVA followed by Sidak's comparisons test ***$P < 0.001$. **f** Chemogenetic inhibition of the LDTg in defeated LDTg[hM4] mice prevents the apparition of VTA DA hyperexcitability (number of cell/mice: Naive/Sal = 21/5; Naive/CNO = 12/4; CSD/Sal = 20/5; CSD/CNO = 18/5). Interaction treatment × current $F(18,384) = 2.39$, $P = 0.001$; repeated-measures two-way ANOVA followed by Sidak's comparisons test ***$P < 0.001$. Representative voltage traces are shown. **g** Increase in VTA DA AMPA-R/NMDA-R ratio after CSD is prevented by chemogenetic inhibition of LDTg (number of cell/mice: Naive/Sal = 10/5; Naive/CNO = 10/5; CSD/Sal = 8/6; CSD/CNO = 9/6). Interaction treatment × stress $F(1,33) = 4.12$, $P < 0.05$; two-way ANOVA followed by Sidak's comparisons test ***$P < 0.001$. Representative current traces in the absence/presence of the NMDA antagonist AP-5. All plots depict mean ± SEM

displayed normal social interaction and sucrose intake (Fig. 3b and Supplementary Fig. 5b, respectively). Behavioral responses were comparable in undefeated LDTg[ChAT-hM4] mice treated with either saline or CNO (Fig. 3b and Supplementary Fig. 5b). Consistent with these results, selective silencing of cholinergic LDTg neurons during CSD prevented stress-induced VTA DA neuron hyperexcitability and increases in AMPA-R/NMDA-R ratio (Fig. 3c and Supplementary Fig. 5c, respectively). We also recorded cholinergic LDTg neurons from LDTg[ChAT-hM4] mice. Chronic CNO administration did not affect the excitability profile of these neurons in naive conditions when compared with saline but prevented CSD-induced hyperexcitability (Supplementary Fig. 5d).

In striking contrast, selective silencing of LDTg glutamatergic neurons during CSD failed to prevent behavioral and cellular adaptations to stress. To reach this observation, we injected AAV-hSyn-DIO-hM4-mcherry in the LDTg of Vglut2-Cre mice (here after LDTg[Vglut2-hM4] mice, Fig. 3d). Defeated LDTg[Vglut2-hM4] mice treated with either saline or CNO exhibited significant social aversion (Fig. 3e) and similar hyperexcitability in VTA DA neuron firing (Fig. 3f). Collectively, the above data indicate that cholinergic, neurons of the LDTg are key orchestrators of maladaptations to chronic stress impinging on VTA DA neurons that ultimately drive social aversion and anhedonic processes. Conversely, silencing of LDTg glutamatergic neurons did not yield stress relief, which may reflect compensation from other glutamatergic inputs to the VTA.

**LDTg[→VTA] cholinergic pathway activation gates stress impact.** Due to the widespread projections of LDTg neurons, we wanted to narrow down the observed effect to LDTg projections to the VTA. Also, if activation of these projections is required for stress outcomes, this would imply that increasing activity of LDTg neurons that project to the VTA could favor stress-induced maladaptations. This would result in an enhanced sensitivity of the DA system to a moderate social stress, and the appearance of social aversion. In order to selectively activate LDTg neurons that project to the VTA, we injected a retrograde CAV-2-Cre in the VTA and AAV-hSyn-DIO-hM3-mcherry in the LDTg of WT mice (hereafter LDTg[hM3→VTA] mice, Fig. 4a). This approach allows activation of transduced LDTg neurons in the presence of CNO (Supplementary Fig. 2b). To mimic a moderate social stress, we submitted LDTg[hM3→VTA] mice to a subthreshold defeat

(SubSD) paradigm (Fig. 4a). We and others have previously used this protocol to evaluate the stress-potentiating effects of either optogenetic or pharmacological manipulation of VTA DA neurons[9,21]. We observed that SubSD triggered social aversion in mice receiving CNO, whereas it was ineffective in saline-injected mice. In view of our above results, we hypothesized that this effect could be mediated by Designer Receptor Exclusively Activated by Designer Drug (DREADD)-induced activation of cholinergic inputs from the LDTg to the VTA, via activation of neuronal nicotinic acetylcholine receptors (nAChRs) that are key regulators of VTA DA neurons[21,22]. To test this, we locally infused the non-selective nAChRs antagonist mecamylamine in the VTA, before LDTg[hM3→VTA] mice entered the SubSD protocol (Fig. 4b). This treatment fully prevented the appearance of social aversion resulting from a mild stress combined with DREADD activation of the LDTg→VTA pathway (Fig. 4b). In line with the behavioral data, LDTg[hM3→VTA] mice submitted to SubSD showed enhanced excitability of VTA DA neurons when injected with CNO but not saline, an effect that could be reversed by systemic injection of mecamylamine (Fig. 4c). This set of data indicates that direct LDTg stimulation of VTA neurons paired with a short-lived stress is sufficient to promote behavioral and cellular adaptations, and that this effect requires activation of VTA nAChRs. To unambiguously demonstrate the involvement of LDTg cholinergic neurons projecting to the VTA in this process, we locally infused CNO in the VTA of LDTg[ChAT-hM3] mice to selectively activate cholinergic terminals. Although vehicle-treated mice submitted to SubSD showed normal social interaction, VTA infusion of CNO combined with SubSD triggered significant social aversion (Fig. 4d). Therefore, our data demonstrate that LDTg cholinergic inputs to the VTA drive maladaptations to social stress. It also demonstrates that we can exert a bidirectional control over LDTg neurons to either favor or block adaptations to stress within VTA DA neurons.

**CRF signaling affect LDTg cholinergic neurons.** What is the molecular determinant of stress' action that drives hyperexcitability of cholinergic LDTg neurons? Stressors trigger the activation of the hypothalamo-pituitary adrenal axis, which is initiated by the release of CRF, a neuropeptide that act at hypothalamic and extra-hypothalamic sites[23]. CRF is a potent and fast-acting mediator of endocrine and autonomic responses to stress, and regulates neurotransmission directly through two

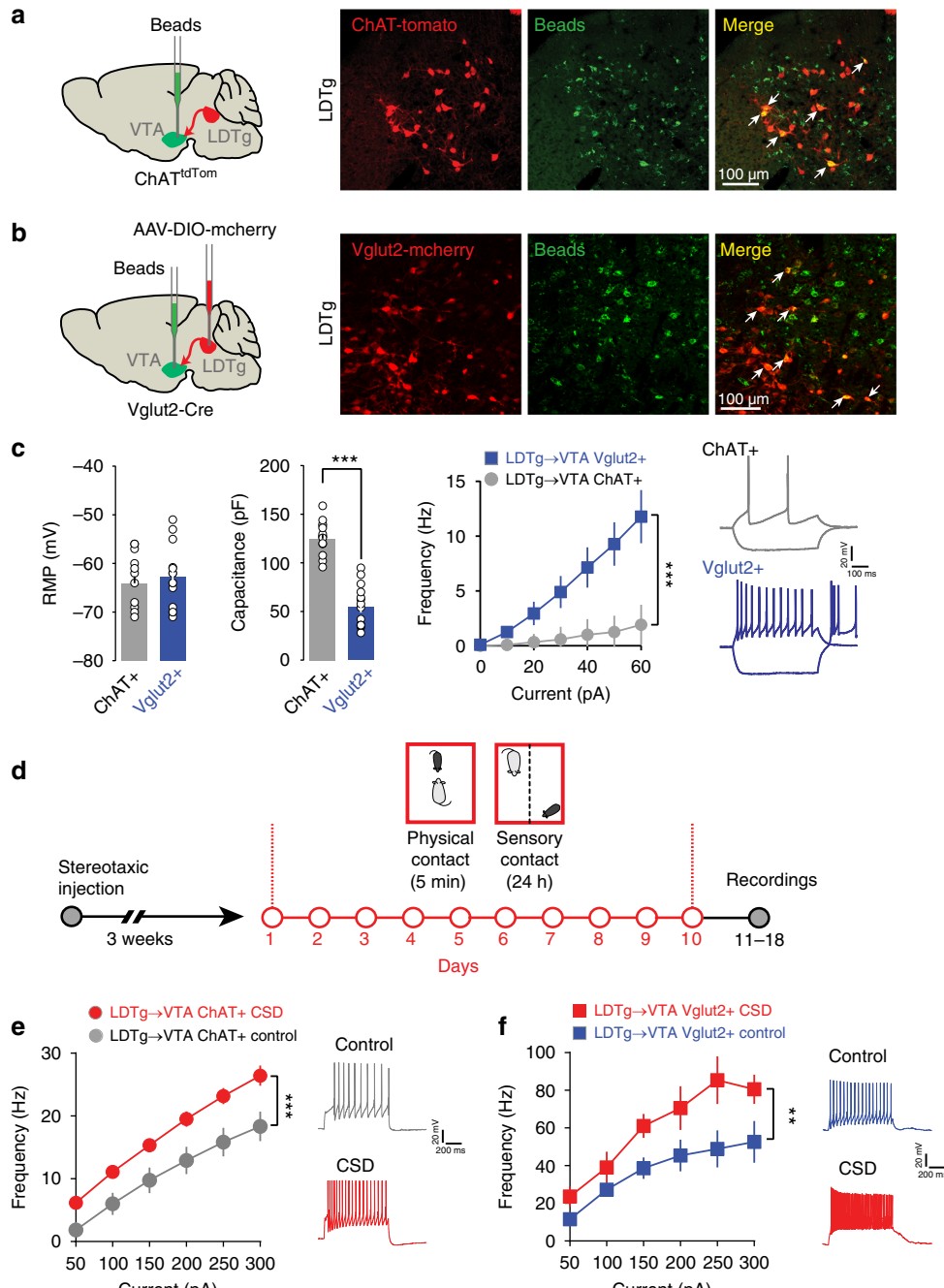

**Fig. 2** Cholinergic and glutamatergic LDTg→VTA neurons are sensitive to chronic stress. **a** ChAT-Cre::tdTomato mice were injected with green retrobeads in the VTA. Confocal images show the co-expression of green beads and tdTomato in cholinergic neurons of the LDTg. **b** Vglut2-Cre mice were injected with an AAV8-hSyn-DIO-mcherry in the LDTg and green retrobeads in the VTA. Confocal images show the co-expression of green beads and mcherry in glutamatergic neurons of the LDTg. **c** Bioelectrical properties of cholinergic and glutamatergic LDTg→VTA neurons. Cholinergic neurons show bigger capacitance and lower firing frequencies at each current step injection (number of cell/mice: ChAT += 15/4; Vglut += 14/3). Interaction cell type × current $F_{(6, 162)} = 15.77$, $P < 0.0001$; repeated-measures two-way ANOVA followed by Sidak's comparisons test ***$P < 0.001$. Representative voltage traces showing response to current injection ($-40$ and $+60$ pA) in both neuronal populations. **d** Schematic experimental time line to assess excitability in cholinergic and glutamatergic neurons after CSD. **e** Patch-clamp recordings in cholinergic LDTg→VTA neurons revealed increased excitability after CSD (number of cell/mice: ChAT + /Naive = 15/3; ChAT + /CSD = 15/4). Interaction treatment × current $F_{(6, 168)} = 19.23$, $P = 0.0001$; repeated-measures two-way ANOVA followed by Sidak's comparisons test ***$P < 0.001$. Representative voltage traces to a 300 pA current injection. **f** Patch-clamp recordings in glutamatergic LDTg→VTA neurons also shows increased excitability after CSD (number of cell/mice: Vglut + /Naive = 13/4; Vglut/CSD = 9/4). Interaction treatment × current $F_{(18, 384)} = 2.39$, $P = 0.001$; repeated-measures two-way ANOVA followed by Sidak's comparisons test, **$P < 0.01$. Representative voltage traces to a 50 pA current injection. All plots depict mean ± SEM

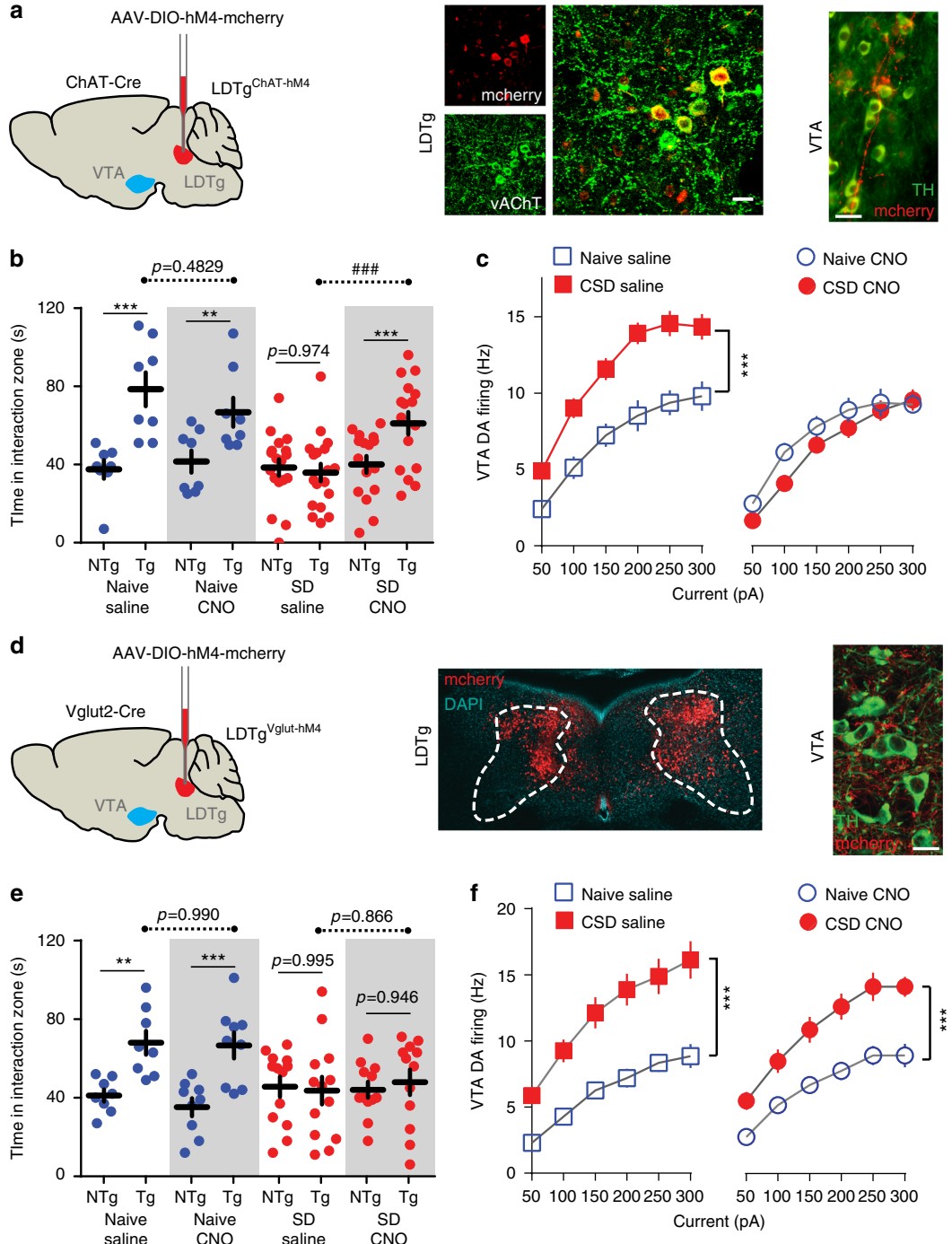

receptor subtypes, CRF receptor 1 (CRF-R1) and CRF-R2[23]. It is therefore possible that the CRF system has a role in social stress processing in the LDTg consequently affecting its output connectivity. A single defeat session in naive mice led to a significant increase (1.8-fold) in plasma CRF levels (basal, 399.9 ± 116.7 pg/ml; acute aggression, 723.5 ± 178.6 pg/ml; $n = 8–12$ per group, $P < 0.05$, Mann–Whitney). To ascertain the functional role of stress-induced CRF release onto LDTg cholinergic neurons, and the respective contribution of CRF-1 or CRF-2 receptor subtypes, we performed whole-cell patch-clamp recordings in slices from ChAT[tdTom] mice (Fig. 5a). In current-clamp mode and after reaching a stable baseline, the CRF-R1 agonist stressin I (1 μM) or the CRF-R2 agonist urocortin III (1 μM) were applied to the perfusion system. Stressin I application, but not urocortin III,

consistently initiated action potential discharge in all cholinergic neurons tested, and this effect disappeared after washout of the drug (Fig. 5a). In contrast, stressin I and urocortin III failed to elicit any notable responses in LDTg glutamatergic neurons (Fig. 5b). To further refute the involvement of CRF-R1, a subset of LDTg glutamatergic neurons were depolarized to − 55 mV by current injection before applying the agonist. Even in this condition, stressin I failed to initiate action potential firing (five cells from two mice). These data suggest that after CSD, cholinergic neurons of the LDTg could be overly activated by the release of CRF via activation of excitatory CRF-R1. To answer this question, WT mice were subjected to CSD where each defeat session was preceded by a local bilateral injection of a CRF-R1-specific antagonist, CP376395 (500 ng/site), or saline. Control mice

**Fig. 3** Selective chemogenetic inhibition of LDTg cholinergic but not glutamatergic neurons during CSD is sufficient to prevent depressive behaviors and VTA DA dysregulation. **a** Bilateral stereotaxic injection of AAV8-hSyn-DIO-hM4-mcherry in the LDTg of ChAT-Cre mice (LDTg[ChAT-hM4] mice). Confocal images showing colocalization of mcherry and vesicular acetylcholinesterase transporter (vAChT) Scale bar = 20 μm. Epifluorescence micrograph showing cholinergic fibers expressing mcherry in close interaction with tyrosine hydroxylase (TH)-positive VTA dopamine neurons. Scale bar = 25 μm. **b** Social interaction times in undefeated or defeated LDTg[ChAT-hM4] mice treated with saline or CNO. Saline-injected defeated mice showed marked social aversion, whereas CNO injection in the defeated group restored social interaction (number of mice: Naive/Sal = 8; Naive/CNO = 8; CSD/Sal = 18; CSD/CNO = 16). Interaction treatment × target $F(3,46)$ = 9.62, $P$ = 0.0001; Interaction treatment × stress $F(1,46)$ = 7.99, $P$ < 0.01; repeated-measures two-way ANOVA followed by Sidak's comparisons test, asterisks depict NTg vs. Tg comparisons, hash depicts comparisons between saline vs. CNO in the Tg condition; **$P$ < 0.01, ***$P$ < 0.001, ###$P$ < 0.001. **c** Inhibition of cholinergic LDTg neurons during CSD prevented increased excitability in VTA DA neurons (number of cell/mice: Naive/Sal = 11/4; Naive/CNO = 13/4; CSD/Sal = 22/5; CSD/CNO = 18/5). Interaction treatment × current $F(18,360)$ = 4.66, $P$ = 0.0001; repeated-measures two-way ANOVA followed by Sidak's comparisons test ***$P$ < 0.001. **d** Bilateral stereotaxic LDTg injection of AAV8-hSyn-DIO-hM4-mcherry of Vglut2-Cre mice (LDTg[Vglut2-hM4] mice). Confocal image shows representative injection site. Confocal image showing glutamatergic fibers expressing mcherry in close interaction with TH-positive VTA dopamine neurons. Scale bar = 20 μm. **e** Social interaction times in undefeated or defeated LDTg[Vglut2-hM4] mice treated with saline or CNO. Silencing of LDTg glutamatergic neurons by administration of CNO during stress did not prevent the appearance of social aversion (number of mice: Naive/Sal = 8; Naive/CNO = 9; CSD/Sal = 13; CSD/CNO = 12). Interaction treatment × target $F(3,38)$ = 6.80, $P$ = 0.0009; Interaction treatment × stress $F(1,38)$ = 0.1619, $P$ = 0.6897; repeated-measures two-way ANOVA followed by Sidak's comparisons test, asterisks depict NTg vs. Tg comparisons, hash depicts comparisons between saline vs. CNO in the Tg condition; **$P$ < 0.01, ***$P$ < 0.001, ###$P$ < 0.001. **f** Patch-clamp recordings in brain slices showed that inhibition of glutamatergic LDTg neurons during CSD did not prevent increased excitability in VTA DA neurons (number of cell/mice per condition: Naive/Sal = 12/4; Naive/CNO = 14/4; CSD/Sal = 10/5; CSD/CNO = 10/5). Interaction treatment × current $F(6,108)$ = 0.95, $P$ = 0.466; repeated-measures two-way ANOVA followed by Sidak's comparisons test ***$P$ < 0.001). All plots depict mean ± SEM

received the same treatment but without defeat (Fig. 5b) and animals were tested in drug-free conditions. Chronic infusion of CP376395 did not affect social approach or sucrose preference in undefeated mice; however, CP376395 fully prevented the appearance of stress-induced social aversion and anhedonia (Fig. 5c, d). We then assessed the state of VTA DA neuron excitability and, in accordance with the behavioral results, local CP376395 infusion in the LDTg prevented hyperexcitability induced by CSD (Fig. 5e). This demonstrates the ability of CRF signaling in the LDTg to promote depressive-like behaviors via dysregulation of VTA DA function.

## Discussion

Here we demonstrated that chronic social stress exposure produces profound dysregulation of excitatory inputs from the LDTg to the VTA via CRF signaling. Selective inhibition of cholinergic, but not glutamatergic, LDTg neurons prevents stress-induced cellular adaptations within VTA DA neurons and the appearance of anhedonia and social withdrawal.

Central cholinergic neurotransmission is a powerful neuromodulator that, in physiological conditions, affects neuronal excitability and synaptic strength to synchronize neuronal networks, in order to guide adaptive behaviors[24,25]. Dysregulation of acetylcholine tone in discrete brain areas can lead to pathophysiological states[26,27], and in particular a heightened cholinergic transmission has long been postulated to contribute to depression[28]. This hypothesis is supported by the observations that depressive symptomatology can be induced by cholinomimetics or inhibition of acetylcholinesterase[29–31]. Despite this evidence, the neuronal circuit underlying depression, as well as the cellular and synaptic changes driven by acetylcholine, are still poorly understood. Our data uncover a cardinal role of cholinergic neurons of the LDTg in the appearance of chronic stress-induced depressive-like behavioral manifestations. We show that CSD increased the excitability of LDTg cholinergic neurons that project to the VTA. Increased cholinergic tone would favor DA neurons firing by two mechanisms. Activation of excitatory postsynaptic nAChRs produces depolarization and firing activity of DA neurons[32]. In addition to direct activation, acetylcholine release can enhance the release of glutamate from other inputs via activation of presynaptic nAChRs[33,34] (Fig. 6). We put in evidence the importance of this cholinergic/nicotinic mechanism by pharmacological blockade of VTA nAChRs. This is in line with our previous data showing that constitutive ablation of nAChRs is sufficient to prevent CSD-induced hyperactivity of DA neurons[21].

Although the mechanisms underlying CSD-induced increases in DA firing are at present unclear, our study suggests that adaptations in both intrinsic and synaptic plasticity could contribute to this process[35]. For example, we showed that CSD induces changes in AMPA-R/NMDA-R ratio, a proxy for glutamatergic synaptic strength. Our data suggest that cholinergic LDTg inputs may serve as a permissive signal. Indeed, silencing of the LDTg cholinergic pathway was effective in preventing stress-induced changes in activity and glutamatergic plasticity of DA neurons. Unlike cholinergic neurons, silencing of LDTg glutamatergic neurons did not produce stress relief. This may be explained by the fact that the VTA receives numerous sources of glutamatergic inputs[12,36], which have been shown to be strengthened by aversive stimuli[37]. It is evident that the glutamatergic synapse undergoes significant remodeling that are likely to contribute to deregulation of VTA DA neurons activity but the molecular mechanisms and input sources affected by stress are yet to be elucidated.

VTA DA neurons not only respond to rewarding stimuli but they can also adapt to aversive and stressful events[5]. Inputs to the VTA likely have a role in shaping stress-induced depressive-like behaviors. For example, a recent study shows that pallidal parvalbumin neurons adapt their activity in response to chronic stress, affecting VTA DA excitatory/inhibitory balance[38]. In another study, noradrenergic inputs from the locus coeruleus were shown to promote stress resilience putatively through an inhibitory mechanism[39]. The LDTg to VTA pathway has been studied for its role in reward. For example, direct optogenetic-mediated activation of LDTg terminals in the VTA is sufficient to produce conditioned place preference[16] and a similar effect is achieved when the stimulation is restricted to cholinergic terminals[40]. Our study addresses the role of this pathway in stress processing and demonstrates that aversive encounters with a dominant male can produce activation of LDTg neurons. Importantly, chemogenetic activation of the LDTg to VTA pathway paired with a mild social stress promotes the development of depressive-like symptoms and the aberrant increase in VTA DA neurons activity. These results indicate that LDTg

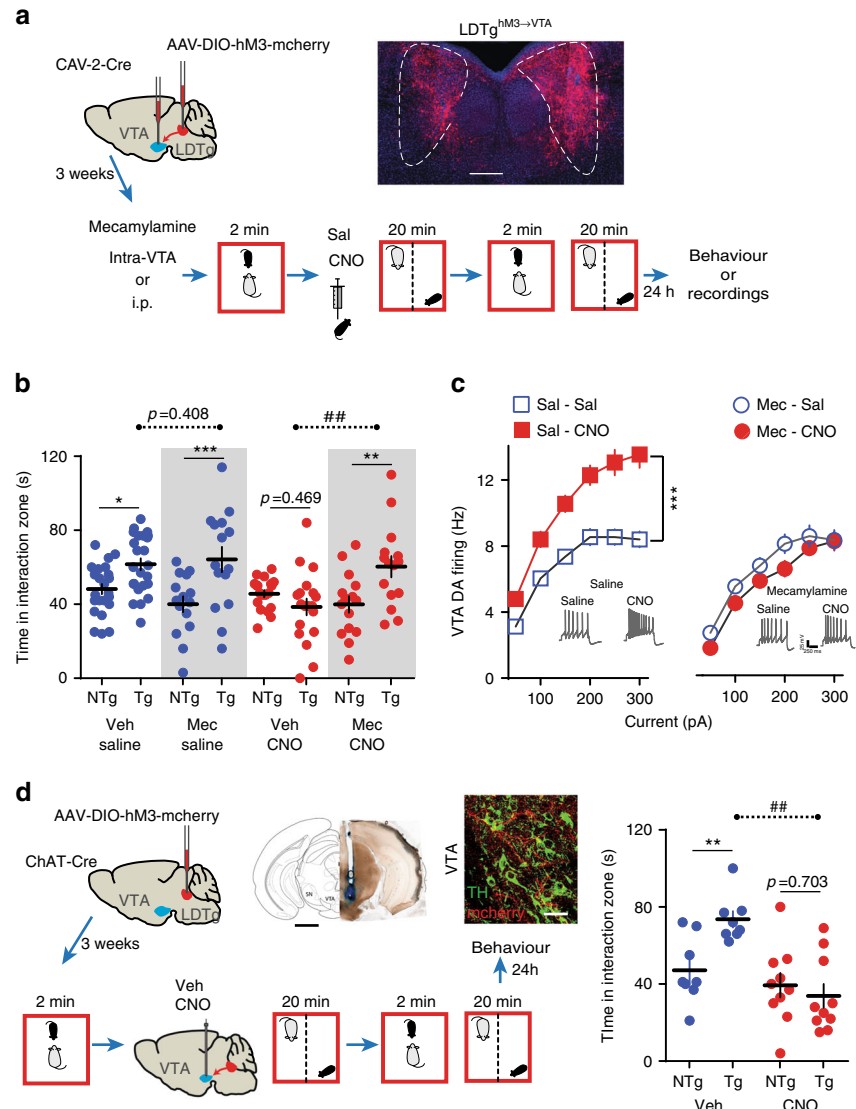

**Fig. 4** LDTg cholinergic inputs to the VTA promote depressive-like maladaptations. **a** Bilateral stereotaxic injection of a retrograde CAV-2-Cre in the VTA combined with a bilateral injection of an AAV8-hSyn-DIO-hM3-mcherry in the LDTg of wild-type mice (LDTg^hM3→VTA). Representative image depicting hM3-mCherry expression in LDTg neurons projecting to the VTA and schematic experimental time line. Mice were implanted with cannula guides over the VTA, allowing for local infusion. **b** A subthreshold defeat (SubSD) exposure combined with CNO-induced activation of LDTg neurons projecting to the VTA elicits social aversion. Mecamylamine (Mec; 1 μg/site) infusion in the VTA before SubSD is sufficient to prevent this effect (number of mice: Vehicle/Sal = 24; Mec/Sal = 15; Vehicle/CNO = 19; Mec/CNO = 15). Interaction treatment × target F(3,69) = 7.69, P = 0.0002; Interaction treatment × stress F(1,69) = 0.1564, P = 0.2156; repeated-measures two-way ANOVA followed by Sidak's comparisons test, asterisks depict NTg vs. Tg comparisons, hash depicts comparisons between saline vs. CNO in the Tg condition; *P < 0.05, **P < 0.01, ***P < 0.001, ##P < 0.01. **c** A different cohort of LDTg^hM3→VTA mice received either systemic saline or mecamylamine (1 mg/kg) before being submitted to SubSD. CNO-induced activation of LDTg neurons projecting to the VTA combined with SubSD produced hyperactivity of VTA DA neurons. This effect was fully prevented by mecamylamine. Number of cell/mice: Sal/Sal = 18/4; Sal/CNO = 18/5; Mec/Sal = 12/4; Mec/CNO = 16/5. Interaction treatment × current F(18,360) = 6.58, P = 0.0001; repeated-measures two-way ANOVA followed by Sidak's comparisons test ***P < 0.001. Representative voltage traces from each experimental condition are shown. **d** ChAT-Cre mice were injected with an AAV8-hSyn-DIO-hM3-mcherry in the LDTg and guides were implanted bilaterally over the VTA. Representative image shows cannula placement, scale bar = 1.2 mm. Confocal image showing cholinergic fibers expressing mcherry in close interaction with TH-positive VTA dopamine neurons. Scale bar = 30 μm. Local VTA infusion of CNO combined with a SubSD resulted in strong social aversion, an effect not seen in vehicle-treated mice (number of mice per condition: Vehicle = 8; CNO = 10). Interaction F(1,16) = 8.83, P = 0.009; repeated-measures two-way ANOVA followed by Sidak's comparisons test, asterisks depict NTg vs. Tg comparisons, hash depicts comparisons between saline vs. CNO in the Tg condition; **P < 0.01, ##P < 0.01. All plots depict mean ± SEM

projections to the mesolimbic pathway not only convey rewarding information but also promote alertness in cases of threatening situations. The LDTg is therefore a more versatile brain structure than originally thought and should therefore be considered as a key upstream regulator of the VTA that shapes responses to salient stimuli, either rewarding or aversive.

There is at present substantial evidence for the involvement of the DA system in the pathophysiology of depressive-like disorders[4]. Most preclinical models of depression rely on exposure to stressful experiences[41]. Despite inducing shared behavioral maladaptations, these models are underpinned by different, and sometimes opposite, cellular changes within the DA system.

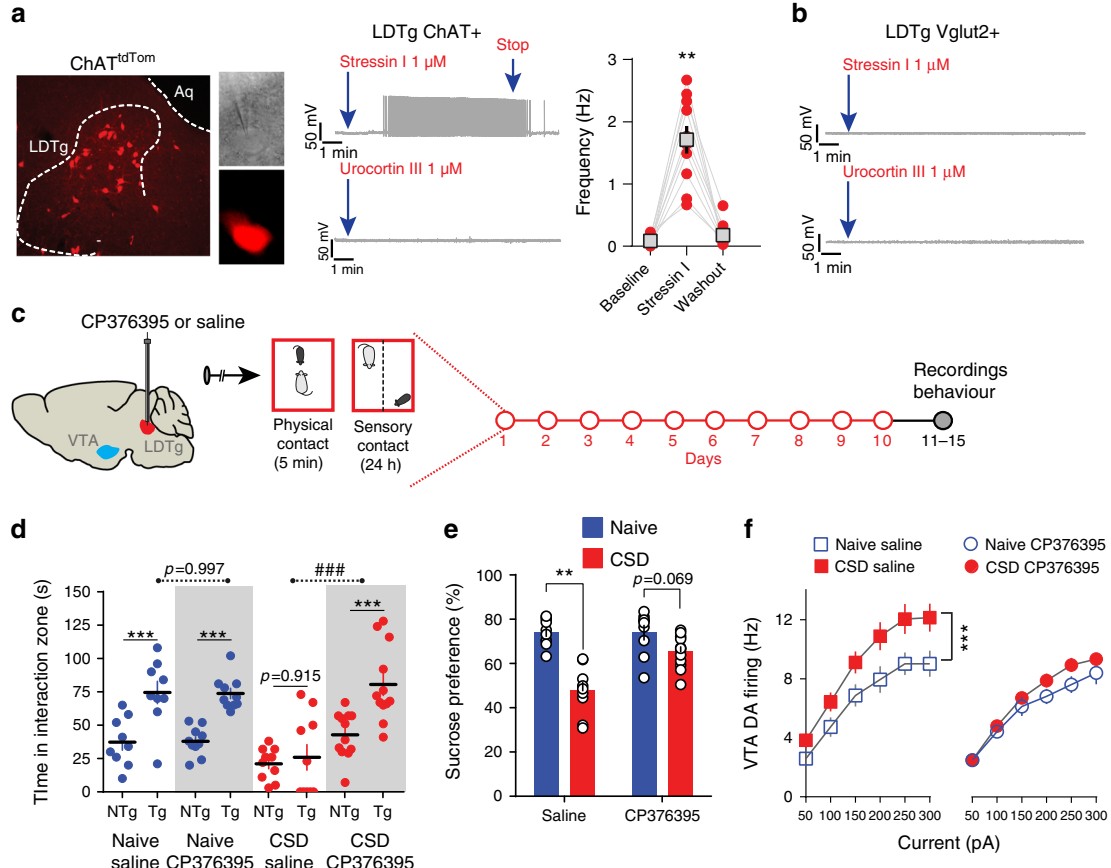

**Fig. 5** CRF-1 receptors expressed in cholinergic LDTg neurons mediate stress-induced depressive-like behaviors and dysregulation of VTA DA neurons. **a** Confocal image depicting the LDTg in a ChAT-Cre::tdTtomato (ChAT$^{tdTom}$) mouse coronal section. Visualization of tdTomato-tagged cholinergic neurons in the electrophysiological set-up during patch-clamp recordings. In current-clamp mode, most cholinergic neurons were silent, but bath application of the CRF receptor 1 agonist stressin I initiated action potential discharge. This effect was not seen with bath application of the CRF receptor 2 agonist urocortin III (number of cell/mice = 7/3). Quantification of the firing frequency in the absence and presence of stressin I (number of cell/mice = 10/4, **$P < 0.01$, repeated-measures one-way ANOVA). **b** To visualize LDTg glutamatergic neurons, Vglut2-Cre mice were injected with an AAV8-hSyn-DIO-mcherry. Representative current-clamp recordings showing that stressin I (number of cell/mice = 9/3) or urocortin III (number of cell/mice = 9/3) failed to elicit discharge activity in this neuronal population. **c** Schematic experimental time line. Wild-type mice were implanted with cannula guides over the LDTg, allowing for local infusion before each social defeat session. **d** Local LDTg infusion of the CRF-1 antagonist CP376395 before defeat prevented the appearance of social aversion (number of mice: Naive/Vehicle = 9; Naive/CP376395 = 10; CSD/Vehicle = 10; CSD/ CP376395 = 12). Interaction treatment × target F(3,37) = 6.33, $P = 0.0014$; Interaction treatment × target F(1,69) = 7.69, $P = 0.0002$; Interaction treatment × stress F(1,37) = 12.23, $P = 0.0012$; repeated-measures two-way ANOVA followed by Sidak's comparisons test, asterisks depict NTg vs. Tg comparisons, hash depicts comparisons between saline vs. CNO in the Tg condition; ***$P < 0.001$, ###$P < 0.001$. Anatomical coordinates and maps were adjusted from Paxinos[48]. **e** Sucrose preference was absent in saline-treated defeated mice, but infusion of CP376395 in the LDTg during CSD restored sucrose consumption (number of mice: Naive/Vehicle = 9; Naive/CP376395 = 10; CSD/Vehicle = 10; CSD/CP376395 = 11). Interaction treatment × stress F(1,36) = 10.63, $P = 0.0024$; two-way ANOVA followed by Sidak's comparisons test ***$P < 0.001$. **f** Patch-clamp recordings in brain slices showed that increased excitability to current injection in VTA DA neurons was observed in saline-treated but not in CP376395-treated mice (number of cell/mice: Naive/Sal = 14/5; Naive/CP376395 = 11/4; CSD/Sal = 14/5; CSD/CP376395 = 19/5). Interaction current × treatment F(18,324) = 3.19, $P < 0.0001$; repeated-measures two-way ANOVA followed by Sidak's comparisons test, ***$P < 0.001$. All plots depict mean ± SEM

Indeed, the two most widely used mouse models of depression, the unpredictable chronic mild stress (UCMS), and the CSD produce decreases[42,43] or increases[8,9,11] in VTA DA neurons' activity, respectively. Presently, the nature of these discrepancies is not clear and may be related to differences in the duration of stress (4–6 weeks for UCMS vs. 10 days for CSD) or protocol (unpredictable stress vs. contextual association with an aggressor). Importantly, in both models, antidepressant-like effects can be achieved by phasic optogenetic stimulation of VTA DA neurons[42,44], likely restoring cellular homeostasis and synaptic flexibility within the system.

In the last decade, technological advances have allowed scientists to modulate specific cellular pathways in freely moving animals, in order to dissect brain functions. Among these, chemogenetic (DREADD) approaches, which combine the use of modified G-coupled protein receptors that are activated by CNO, have become increasingly popular[18]. A recent publication by Gomez et al.[45] raised potential caveats due to the back-metabolism of CNO into clozapine in vivo. Nevertheless, these findings do not discount our conclusions since we have shown in non-DREADD-expressing mice that CNO per se did not impinge on two classical patterns of activity of VTA DA neurons in vivo (slow, single-spike firing, and fast-bursting activity) and did not alter behavioral and cellular adaptations to CSD. Moreover, given that we did not observe stress relief when inhibiting LDTg glutamatergic neurons further reinforces the absence of unwanted CNO side effects in this paradigm.

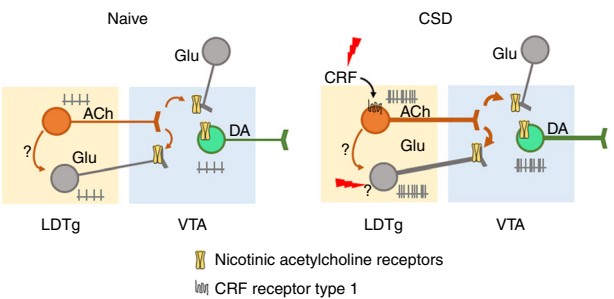

**Fig. 6** Schematic model of the findings. In naive conditions, LDTg excitatory projections regulate the firing and bursting activity of VTA DA neurons. Chronic social defeat (CSD) increases the excitability of LDTg cholinergic and glutamatergic neurons that project to the VTA. Enhanced acetylcholine release increases VTA DA firing via direct activation of neuronal acetylcholine nicotinic receptors, but also by enhancing glutamatergic release via presynaptic modulation. CRF binds to CRF-1 receptors in cholinergic but not glutamatergic neurons, to promote firing in response to stress. The mechanisms by which LDTg glutamatergic neurons increase excitability after CSD are not clear, but may involve local cholinergic innervation, direct stress-related signaling, or others inputs to LDTg glutamatergic neurons

In response to stressful situations, CRF initiates neuroendocrine changes to orchestrate rapid bodily responses to ensure adequate behavioral endpoints[23]. However, due to the broad central distribution of CRF receptors, the precise site(s) of action and the CRF-sensitive cell populations need to be identified. More than 90% of cholinergic neurons in the LDTg express CRF-R1[46]. In accordance, we show here that acute activation of CRF-R1, but not CRF-R2, markedly increased firing of cholinergic LDTg neurons. This is likely to be the molecular event by which chronic stress triggers hyperactivity of cholinergic LDTg neurons. Indeed, we report that locally antagonizing CRF-R1 in the LDTg is sufficient to reverse cellular and behavioral marks of stress. Novel therapies involving modulation of brain pathways are showing encouraging results so far[47]. However, future advances in these therapies will rely on accurate identification of cell-to-cell miscommunication within delineated neural circuits, and its involvement in pathological mental states. Our data point to a dysregulation of a tegmental cholinergic pathway arising to the VTA as a cardinal contributor to stress-induced depressive-like behaviors (Fig. 6). This imbalance could be targeted by either pharmacological antagonism to CRF receptors or, in the future, with novel selective techniques of brain stimulation.

## Methods

**Animals**. All procedures were in accordance with the recommendations of the European Commission (2010/63/EU) for care and use of laboratory animals, and approved by the French National Ethical Committee. We used only male C57BL/6J (Janvier Labs, France), choline acetyltransferase ChAT-IRES-Cre Knock-In mice (ChAT-Cre mice, The Jackson Laboratory, stock number: 006410) and vglut2-Cre mice[20]. To specifically label cholinergic neurons, ChAT-Cre mice were crossed with tdTomato reporter line (The Jackson Laboratory, stock number 007909). Experimental transgenic animals were heterozygous and backcrossed on a C57BL/6J background.

**Reagents**. Clozapine N-oxyde was purchased from Enzo Life (France), Stressin I, mecamylamine and CP376395 from Tocris Cookson (UK), Urocortin III from Sigma-Aldrich (France), and Xylazine/Ketamine from Centravet (France). All drugs for in vivo administration were diluted in saline (0.9% NaCl). For bath application of drugs on brain slices, dilution was done in aCSF (artificial cerebrospinal fluid).

**Stereotaxic injections and cannula implantation**. Stereotaxic injections were performed using a stereotaxic frame (Kopf Instruments) under general anesthesia with xylaizne and ketamine (10 mg/kg and 150 mg/kg, respectively). Anatomical coordinates and maps were adjusted from Paxinos[48]. The injection rate was set at

100 nl/min. To identify LDTg neurons projecting to the VTA, mice (5 weeks old) were injected with green fluorescent retrobeads (200 nl/site; LumaFluor, Inc.) bilaterally in the VTA (AP: − 3.2 mm, ML: ± 0.6 mm, DV: − 3.7 mm from the bregma). AAV8-hSyn-hM4D-mCherry and hSyn-DIO-hM4D-mCherry (titration 10$^{12}$ particles/ml) were purchased from the U.N.C. (University of North Carolina, USA) vector core and bilaterally injected (300 nl/site) in the LDTg (anteroposterior AP: − 4.7 mm, mediolateral ML: ± 0.5 mm, dorsoventral DV: − 3.6 mm from the bregma) of either WT, ChAT-Cre, or Vglut2-Cre mice (5 weeks old). Animals were given a 3 weeks' recovery period to allow sufficient viral expression.

For projection-specific manipulation, WT mice (5 weeks old) were injected with CAV-2-Cre (200 nl/site) in the VTA and with hSyn-DIO-hM3D-mCherry (300 nl/site) in the LDTg (hereafter named LDTg$^{hM3→VTA}$ mice). We thank E.J. Kremer and the Plateforme de Vectorologie de Montpellier for providing CAV-2-Cre. Two weeks later, LDTg$^{hM3→VTA}$ mice were implanted with double-guide cannulas above the VTA for local injections of vehicle or mecamylamine. Double-guide cannulas (ref. C235G, Plastic One), containing dummies (ref. C235DC, Plastic One) and protected by a dust cap (ref 303DC, Plastic One) were placed 1 mm above the VTA (AP: − 3.2 mm, ML: ± 0.6 mm, DV: – 3.7 mm from the bregma) and fixed to the skull with four screws and dental cement. LDTg$^{hM3→VTA}$ mice were allowed to recover for a minimum of 1 week. To target the VTA, cannulas were inserted through the guides to allow drug delivery at DV: − 4.7 mm.

For bilateral drug injections in freely moving mice of CP376395, double-guide cannulas containing dummies and protected by a dust cap were placed 1 mm above the LDTg (AP: − 4.7 mm, ML: ± 0.5 mm, DV: − 2.6 mm from the bregma) of 7 weeks old WT mice and fixed to the skull with four screws and dental cement. Mice were allowed to recover for a minimum of 1 week. To target the LDTg, cannulas were inserted through the guides to allow drug delivery at DV: − 3.6 mm.

Drugs or vehicle were injected at flow rate of 100 nl/min. Cannulas were left in place for another 5 min to avoid backflow.

**CSD stress and behavioral tests**. The CSD stress paradigm was performed as previously[8]. WT, transgenic mice, and their respective control littermates were subjected to 10 consecutive days of social defeat by former CD1 breeder male mice. Each experimental mouse faced a new CD1 every day and sessions of defeat were limited to a maximum of 5 min. Mice were then maintained in sensory, but not physical, contacts with the dominant male through a semi-permeable barrier. Undefeated mice (Naive) were not confronted with a dominant male but lived in similar housing conditions, separated by a semi-permeable barrier. For CNO experiments, CNO (1 mg/kg, i.p.) or saline (10 ml/kg, i.p.) were administered 30 min before each defeat session. Undefeated mice were treated accordingly without being defeated. For CP376395 experiments, local infusion within the LDTg of either CP376395 (500 ng/site in a final volume of 300 nl) or saline (300 nl/site) was done at a rate of 100 nl/min. Cannulas were then left in place for 5 min to prevent backflow. For CSD mice, confrontation with the CD1 occurred after an additional 2 min period (i.e., 10 min after starting local drug infusion). Naive animals were returned to their home cage. Cannula placements were confirmed postmortem. All behavioral tests were performed in drug-free conditions.

Social interaction was performed 24 h after the last defeat (day 11) in a low luminosity environment (7 ± 2 lux). Experimental mice were exposed to two consecutive sessions (150 s each) in an open-field containing initially an empty perforated box ("target −" condition), which was then replaced by a box containing an unfamiliar CD1 mouse ("target +" condition). The time spent in the interaction zone surrounding the box was recorded and used as an index of social interaction.

Anhedonia was measured in a two-bottles choice procedure for assessing sucrose preference. Mice were acclimatized to two bottles containing water during the last 3 days of the CSD paradigm. Following the social interaction test, mice were allowed to freely drink from a bottle of water or sucrose (0.5%). Bottles were weighed and rotated daily. Sucrose preference was calculated as a percentage 100 × [volume of sucrose consumed/(volume of sucrose + volume of water consumed)] and was averaged over a period of 3 days.

**Subthreshold social defeat**. The substhreshold social defeat paradigm was based on Morel et al.[21].

LDTg$^{hM3→VTA}$ mice received locally vehicle (saline 300 nl/site) or mecamylamine (1 µg in 300 nl/site)[49] in the VTA with the same technical details as described above for CP376395 experiments. LDTg$^{hM3→VTA}$ mice were then individually introduced into a CD1 cage for 2 min of social defeat. Following this, LDTg$^{hM3→VTA}$ mice received a single saline (10 ml/kg) or CNO (1 mg/ml) injection and were maintained with the CD1 mice separated through a partition for 20 min. The social defeat was repeated once and mice assessed in the social interaction test approximately 24 h later as described for the CSD paradigm. For electrophysiological recordings, LDTg$^{hM3→VTA}$ mice were treated accordingly but received acute saline (10 ml/kg) or mecamylamine (1 mg/kg) i.p. 10 min before the first SubSD episode.

To selectively activate LDTg cholinergic terminals in the VTA, guides were placed bilaterally above the VTA of LDTg$^{ChAT-hM3}$ mice to locally infuse CNO. Local CNO infusion has been recently used for remote control of terminals activity[50,51]. LDTg$^{ChAT-hM3}$ mice were submitted to SubSD as described above. Immediately after the first session of brief social defeat, CNO (10 µM; 300 nl) or vehicle were infused at a rate of 100 nl/min, while mice were still facing the CD1

mouse through the partition preventing physical contacts. Cannulas were left in place for 5 min and mice remained facing the CD1 for another 11 min (i.e., total time of visual and olfactory contacts of 20 min) before entering the second brief social defeat session. Mice were then returned to their home cage and tested for social interaction 24 h later.

**In vitro patch-clamp recordings**. Mice were anesthetized (Ketamine 150 mg/kg/ Xylazine 10 mg/kg) and transcardially perfused with aCSF for slice preparation on days 11–14. For VTA recordings, horizontal 250 μm slices were obtained in bubbled ice-cold 95% $O_2$/5% $CO_2$ aCSF containing (in mM): KCl 2.5, $NaH_2PO_4$ 1.25, $MgSO_4$ 10, $CaCl_2$ 0.5, glucose 11, sucrose 234, $NaHCO_3$ 26. Slices were then incubated in aCSF containing (in mM): NaCl 119, KCl 2.5, $NaH_2PO_4$ 1.25, $MgSO_4$ 1.3, $CaCl_2$ 2.5, $NaHCO_3$ 26, glucose 11, at 37 °C for 1 h, and then kept at room temperature. For LDTg recordings, coronal 250 μm slices were obtained using the same solutions but recovery at 37 °C lasted 15 min.

Slices were transferred and kept at 32–34 °C in a recording chamber superfused with 2.5 ml/min aCSF. Visualized whole-cell voltage-clamp or current-clamp recording techniques were used to measure synaptic responses or excitability, respectively, using an upright microscope (Olympus France). Putative DA neurons were recorded in the lateral VTA and identified using common criteria such as localization, cell body size, broad action potential, and large Ih current[52]. We compared the profile of putative VTA DA neurons with those recorded using the DA reporter mouse line PITX3-GFP[53] yielding identical profiles (Supplementary Fig. 6). As reported, this method might introduce bias and neglect VTA DA neurons with short action potentials[54]; however, previous results have shown that chronic social stress-induced depressive behaviors is mostly associated with maladaptations in DA neurons with broad spikes[44,55], which are more abundant in the lateral part of the VTA[56].

Current-clamp experiments were obtained using a Multiclamp 700B (Molecular Devices, Sunnyvale, CA). Signals were collected and stored using a Digidata 1440 A converter and pCLAMP 10.2 software (Molecular Devices, CA). In all cases, access resistance was monitored by a step of $-10$ mV (0.1 Hz) and experiments were discarded if the access resistance increased more than 20%. Internal solution contained (in mM): K-D-gluconate 135, NaCl 5, $MgCl_2$ 2, HEPES 10, EGTA 0.5, MgATP 2, NaGTP 0.4. Depolarizing (0–300 pA) or hyperpolarizing (0–450 pA) 800 ms current steps were used to assess excitability and membrane properties of LDTg and VTA neurons.

AMPA-R/NMDA-R ratio was assessed in voltage-clamp mode using an internal solution containing (in mM) 130 CsCl, 4 NaCl, 2 $MgCl_2$, 1.1 EGTA, 5 HEPES, 2 $Na_2ATP$, 5 sodium creatine phosphate, 0.6 $Na_3GTP$, and 0.1 spermine. Synaptic currents were evoked by stimuli (60 μs) at 0.1 Hz through a glass pipette placed 200 μm from the patched neurons. Evoked-EPSCs were obtained at $V = +40$ mV in the absence and presence of the NMDA-R antagonist APV as previously described[21]. In all cases, offline analyses were performed using Clampfit 10.2 (Axon Instruments, USA) and Prism (Graphpad, USA).

**In vivo electrophysiological recordings**. Single-unit extracellular recordings of VTA DA cells were performed in anesthetized (chloral hydrate 8%, 400 mg/kg i.p.) mice as described previously[57]. Glass electrodes (0.5% sodium acetate) were lowered in the VTA according to stereotaxic coordinates (AP: $-3$ to $-4$ mm; ML: 0.1 to 0.7 mm; DV: $-4$ to $-4.8$ mm from the bregma). To distinguish DA from non-DA neurons the following parameters were used: (1) firing rate (between 1 and 10 Hz); (2) action potential duration between the beginning and the negative trough superior to 1.1 ms. The spontaneous frequency and bursting activity of DA neurons were compared in naive LDTg[GFP] and LDTg[hM4] mice receiving an acute injection of either saline or CNO. The neuron response to these i.p. injections was determined by the differences between the maximum of fluctuation on a 3 min period before and after injection.

**Immunohistofluorescence**. Mice were deeply anaesthetized with pentobarbital (Centravet, France) and transcardially perfused with cold phosphate buffer (PB: 0.1 M $Na_2HPO_4$/$NaH_2PO_4$, pH 7.4), followed by 4% paraformaldehyde (PFA) in PB 0.1 M. Brains were post-fixed overnight in 4% PFA-PB. Free-floating vibratome sections (50 μm) were obtained, and correct retrobeads and AAVs injections were confirmed for each animal.

To specifically label VTA DA neurons, midbrain sections were incubated (30 min) in PBS-BT (phosphate-buffered saline (PBS), 0.5% bovine serum albumin, 0.1% Triton X-100) with 10% normal goat serum (NGS). Sections were then incubated (4 °C) in PBS-BT, 1% NGS, with mouse anti-tyrosine hydroxylase (1/ 1000; Millipore Cat MAB318 lot 2211927) for 36 h. Sections were rinsed in PBS and incubated (2 h) in goat anti-mouse Alexa488 secondary antibody (1:1000, Vector Laboratories, Burlingame, CA) in PBS-BT, 1% NGS. Sections were rinsed with PBS and incubated 5 min with DAPI before mounting with Moviol. Similar procedures were followed to label c-Fos-, NeuN-, and cholinergic-positive neurons, using primary anti-c-Fos (1:1000, anti-rabbit, Abcam Cat Ab190289 lot GR266619-1), anti-NeuN (1:1000; anti-mouse, Millipore Cat MAB377 lot 2279235), anti-DsRed (1:1000, anti-rabbit, Clontech Cat 632496 lot 632496), anti-mCherry (1:1000, anti chicken, Abcam Cat ab205402 lot AV4500E), and anti-vesicular acetylcholine transporter (1:1000, anti-guinea pig, Chemicon Cat AB1588 lot 52817[58]), respectively. For c-Fos counting, LDTg[ChAT-mCherry] or LDTg[Vglut-mCherry]

mice were killed one hour after an acute social defeat episode (5 min) and brain sections obtained as described above. Double c-Fos/mCherry immunohistofluorescence was conducted and images acquired with an Olympus FV10 confocal microscope. Cell counting was done manually by an experimenter blind to the conditions using Image J.

**Data analysis**. Data are presented as means ± SEM and were analyzed using GraphPad Prism 7. Following a D'Agostino-Pearson's test, determining the normality of the distributions, statistical analyses were carried out using two-way analysis of variance with repeated measures when required. Post hoc Sidak's test was used when appropriate. Statistical significance was set at $P < 0.05$.

## Data availability
The data that support these findings of this study are available from the corresponding author upon reasonable request.

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

## Acknowledgements

This work was supported by the CNRS, Institut Universitaire de France, Fondation pour la Recherche Médicale, Fédération pour la Recherche sur le Cerveau and ANR (ANR-13-JSV4-0004 MECONOS) to J.B., ANR-16 NicoStress and INCa to P.F. and J.B. We thank Drs Borgius and Kiehn (Karolinska Institutet, Sweden) for providing the vglut2-Cre transgenic mice and Dr Mark Ungless for PITX3-GFP mice. We thank Drs Meye (University Medical Center of Utrecht) and Mameli (University of Lausanne) for critical comments on the manuscript.

## Author contributions

S.P.F. performed viral injections and in vitro electrophysiology. S.P.F., J.B., and L.B. conducted behavioral experiments. L.B., T.C., and M.S.R. performed microscopy. F.M. performed viral injection, in vivo electrophysiology, and analyzed data. X.M. performed ELISA experiments. P.F. and H.M. provided intellectual and technical inputs. J.B. designed the project. J.B. and S.P.F. analyzed data and wrote the manuscript. All authors provided feedback on writing the manuscript.

## Additional information

**Competing interests:** The authors declare no competing interests.

