## [Peer Review File · Nature Communications]

Reviewers' comments:

Reviewer #1 (Remarks to the Author):

In this manuscript, the authors investigate the role of the laterodorsal tegmentum (LDTg) in regulating stress-induced changes in behavior and VTA physiology. Using slice electrophysiology, the authors show that VTA-projecting cholinergic, but not glutamatergic, LDTg cells increase their activity following chronic social defeat stress (CSD). Chemogenetic silencing of cholinergic LDTg neuronal cell bodies (not projections) abolishes CSD-induced social avoidance and anhedonia, and blocks the increase in DAergic activity. Pharmacology experiments suggest that the hyperactivity of LDTg cholinergic cells following CSD is due to corticotropin releasing factor binding to the CRF-R1 receptor. Overall, experiments are well done and well controlled, data analysis is done well, and the results will likely be of broad interest. I do have some concerns with language that is inaccurate at times.

Major points:

1. Language in the paper pushes the reader toward interpreting this data as reflective of the function of a direct cholinergic LDTg projection to the VTA. Even the title: "Mesopontine cholinergic inputs to midbrain dopamine neurons drive stress-induced depressive-like behaviors". But the authors have not selectively manipulated this projection at any point in the paper and cannot make this claim:

a. The authors have shown that the cholinergic LDTg projection to the VTA is more excitable following chronic social defeat stress.

b. But they have not shown that inhibiting the cholinergic LDTg projection to the VTA protects against stress-induced behavioral and VTA physiological changes. They have shown that inhibiting cholinergic LDTg neurons does this. These neurons could project to many different places, and these effects have not been shown to be specific to the LDTg-VTA circuit.

c. Some language in the paper is careful and accurate and reflects this distinction, but there are phrases here and there that clearly imply that this specific projection mediates these behavioral and physiological effects. "Our data point to a dysregulation of a tegmental cholinergic pathway arising to the VTA as a cardinal contributor to stress-induced depressive-like behaviors." People are going to leave this paper thinking that the cholinergic projection to DA neurons in the VTA mediates stress susceptibility. The authors should either correct misleading language throughout the paper (and the title), or demonstrate that this specific projection mediates the described behavioral and physiological changes. This could be accomplished via CAV-flp in the VTA and a cre- and flp-dependent DREADD in the LDTg; via pharmacological block of cholinergic transmission in the VTA (with caveats); or by a number of other potential methods.

2. Recent work has cast some doubt on the use of electrophysiological criteria to accurately classify VTA neurons as dopaminergic and non-dopaminergic. Broad action potentials have been used to identify DA neurons, but this criterion has been shown to exclude the majority of neurons identified as DA by other methods (see e.g. Cohen et al., 2012). Importantly, other work has shown that VTA DA neurons with different projection targets have different spike widths (Margolis et al., 2008). For example, DA neurons that project to the amygdala tend to be short. In brief: broad action potentials are likely DA, but this selection criterion

excludes the majority of DA neurons and likely introduces bias. The authors should address these issues when describing their VTA recordings. Note: only a single sentence is given regarding the classification of DAergic recordings, citing a paper (Morel et al. 2017) that is not included in the references list.

3. While the authors provide evidence for the role of LDTg cholinergic neurons influencing VTA DA activity in states of stress, these results are not fully integrated into the more recent literature regarding the effect of chronic social defeat stress on dopaminergic firing. Of note, different stress models have demonstrated conflicting effects on DAergic activity, with chronic mild stress (CMS) exerting a decrease in DAergic activity, as opposed to the CSD-induced increase in activity. For example, see Chang & Grace (2014) Biol. Psychiatry and supplementary figures in Tye et al. (2013) Nature. Conflicting results from these stress paradigms should be considered in the discussion section.

Minor points:

1. Please identify the c-Fos+ neurons in Figure 1A via immunohistochemistry. Are they ChAT+ or vglut2+?
2. Figure 1g: the authors don't give numbers for each condition independently. Please give numbers, and if there are fewer data points for the CNO animals please collect more data until there is equivalent statistical power as for the saline condition. For the CNO naive, in particular, there appear to be many fewer data points. Also, the interaction has technically not reached significance.
3. Figure 4A: please give a time base in panel 3 (stressin/urocortin)
4. The last panel of Figure 4A has cut off lines.
5. "Accordingly, we have shown that constitutive ablation of nAChRs is sufficient to prevent CSD-induced hyperactivity of DA neurons": This should include a citation.
6. (p. 14): Morel et al., 2017 is not in the references list.

Reviewer #2 (Remarks to the Author):

The study by Fernandez et al examines laterodorsal tegmentum (LDTg) input to VTA in stress-induced depressive like behavior. The authors use a multidisciplinary approach including DREADDs, in vivo and slice recordings, neuron subtype transgenic mice, and pharmacology combined with a behavioral model of stress-induced depression that has strong face validity for modeling depression. Overall the authors providing exciting data to demonstrate a role for cholinergic LDTg inputs in mediating the behavioral responses and characteristic VTA DA neuron firing, through CRF mediated mechanisms, in stress susceptible outcomes to chronic social defeat stress. This study would be of wide interest to the Neuroscience community especially those interested in circuit mechanism in depression. While the results are compelling the conclusions could be strengthened further through additional experiments and analysis. See specific comments below.

- 1) Given recent studies demonstrating that CNO can be converted to clozapine it is important to include CNO controls in social defeat. Given that the authors observe no

difference in behavior and DA neuron firing with CNO in the Vglut-hM4 group then this essentially acts as a control for CNO. It would be useful if the authors acknowledge this in the discussion.

2) For the social interaction data in Figures 1d, 3b, 3e, and 4c the data should be analyzed by stress x treatment similar to the sucrose preference data. Please also show a between subjects comparison of saline vs. CNO (or CP376395) for the Target condition rather than comparing each group to its no Target score.

3) In Figure 2E the authors demonstrate that LDTg-VTA Vglut2+ neurons display increased firing in CSDS mice. However, their DREADD experiments in Figure 3E and F target all Vglut2+ LDTg neurons. A more specific targeting of LDTg-VTA Vglut2+ neurons (i.e. CNO application to LDTg Vglut2+ terminals in VTA) could provide a more definitive conclusion that these neurons do not mediate the behavioral and DA neuron firing effects in CSDS mice.

4) Should the authors identify an effect with direct DREADD inhibition of LDTg-VTA Vglut2+ neurons or terminals (see above comment) it would be useful to examine the response of this population to Stressin and Urocortin as was performed in ChAT neurons in Figure 4.

5) To further determine if the restoration of social interaction and sucrose preference, using a CRF-R1 antagonist into LDTg in Figure 4c and 4d, is mediated through the ChAT LDTg-VTA neurons it would be useful to examine neuron excitability in this population as in Figure 2e. Similar analysis in LDTg-VTA Vglut2+ neurons will be important if DREADD inhibition of the projection neuron subtype alters behavior (see comment #3).

6) Minor comment- "neurons hyperexcitability" on page 7 line 3 and page 8 line 13 should be "neuron."

Reviewer #3 (Remarks to the Author):

This manuscript discusses an interesting circuit that may mediate depression phenotypes, especially social avoidance induced by chronic social defeat. It does a nice job of dissecting the circuitry of LDTg inputs to VTA DA neurons in the context of social defeat, however questions remain about how to place these findings within the context of a broader, somewhat controversial literature. More data on the mechanism of how acetylcholine released by the LDTg influences the DA circuit could also increase the impact of the findings, although I don't think it should necessarily be required for publication.

Major Concerns:

1. How much of this data is specific to the social defeat stress paradigm? The manuscript references work by Chaudhury et al showing that social defeat stress specifically can cause increases in VTA DA neuron firing (in VTA neurons projecting to NAc only). However, there is an abundance of work suggesting that VTA DA neuron firing promotes motivational

behavior and indeed a paper published side-by-side with Chaudhury et al – Tye et al. – is not referenced or discussed. It is hard to place this manuscript's findings in the broader context of depression without understanding why the social defeat paradigm seems to differ from other stress/depression paradigms, such as chronic mild stress, in terms of its effects on VTA DA excitability. Additionally, Lammel et al (Nature, 2012) have shown that stimulation of LDTg inputs to VTA induces place preference, so why would inhibition reverse depression symptoms? The authors must discuss these issues and limit their conclusions and interpretation accordingly.

2. What are the expression patterns of CRF1 and CRF2 receptors in LDTg?

3. Given recent findings that clozapine is in fact the active, BBB-crossing metabolite of CNO that mediates chemogenetic behavioral effects, and that clozapine acts as a dopamine antagonist, I would like to see more data on the effects of CNO injection alone on in vivo VTA firing patterns and behavior in CSD vs naïve mice. The authors do the proper control of injecting CNO in naïve mice in behavioral experiments, however, I don't see injections of CNO in non-hM4D-expressing mice, which would aid in the interpretation of their results. This could be especially key for the interpretation of sucrose preference in Fig 1e, which does not look like nearly as robust an effect as for social interaction time, and perhaps for interpreting the lack of effect when manipulating LDTg glutamatergic neurons.

4. The connection between VTA DA hyperexcitability and cholinergic inputs from LDTg remains a bit confusing. The authors do acknowledge in the conclusion that more work needs to be done to establish the site of glutamatergic synaptic plasticity onto DA neurons and the relationship between synaptic and intrinsic plasticity, which I think is understandable. A summary schematic could be helpful to illustrate both the conclusions and areas for further study.

5. The authors state "We have shown that constitutive ablation of nAChRs is sufficient to prevent CSD-induced hyperactivity of DA neurons," but is there a reference for this statement? Was the ablation of nAChRs expressed by DA neurons specifically?

Reviewer #1 (Remarks to the Author):

In this manuscript, the authors investigate the role of the laterodorsal tegmentum (LDTg) in regulating stress-induced changes in behavior and VTA physiology. Using slice electrophysiology, the authors show that VTA-projecting cholinergic, but not glutamatergic, LDTg cells increase their activity following chronic social defeat stress (CSD). Chemogenetic silencing of cholinergic LDTg neuronal cell bodies (not projections) abolishes CSD-induced social avoidance and anhedonia, and blocks the increase in DAergic activity. Pharmacology experiments suggest that the hyperactivity of LDTg cholinergic cells following CSD is due to corticotropin releasing factor binding to the CRF-R1 receptor. Overall, experiments are well done and well controlled, data analysis is done well, and the results will likely be of broad interest. I do have some concerns with language that is inaccurate at times.

• Major points:

1. Language in the paper pushes the reader toward interpreting this data as reflective of the function of a direct cholinergic LDTg projection to the VTA. Even the title: “Mesopontine cholinergic inputs to midbrain dopamine neurons drive stress-induced depressive-like behaviors”. But the authors have not selectively manipulated this projection at any point in the paper and cannot make this claim:

a. The authors have shown that the cholinergic LDTg projection to the VTA is more excitable following chronic social defeat stress.

b. But they have not shown that inhibiting the cholinergic LDTg projection to the VTA protects against stress-induced behavioral and VTA physiological changes. They have shown that inhibiting cholinergic LDTg neurons does this. These neurons could project to many different places, and these effects have not been shown to be specific to the LDTg-VTA circuit.

c. Some language in the paper is careful and accurate and reflects this distinction, but there are phrases here and there that clearly imply that this specific projection mediates these behavioral and physiological effects. “Our data point to a dysregulation of a tegmental cholinergic pathway arising to the VTA as a cardinal contributor to stress-induced depressive-like behaviors.” People are going to leave this paper thinking that the cholinergic projection to DA neurons in the VTA mediates stress susceptibility. The authors should either correct misleading language throughout the paper (and the title), or demonstrate that this specific projection mediates the described behavioral and physiological changes. This could be accomplished via CAV-flp in the VTA and a cre- and flp-dependent DREADD in the LDTg; via pharmacological block of cholinergic transmission in the VTA (with caveats); or by a number of other potential methods.

We thank the referee for her/his comments and performed new experiments to address this issue. We performed projection-specific manipulation of LDTg→VTA pathway by injecting CAV-2-Cre in the VTA and AAV-hSyn-DIO-hM3D-mcherry in the LDTg. This approach allows activation of the two cell types of interest for this study, namely cholinergic and glutamatergic, which project to the VTA. We then combined a chemogenetic activation of this pathway with a subthreshold social defeat (SubSD) paradigm. This double hit strategy results in the appearance of social aversion, not observed in SubSD alone. This effect was prevented by VTA local injection of mecamylamine, a general nicotinic receptor antagonist. We mirrored these results by showing that this double hit strategy elicits increased excitability of VTA DA neurons *ex vivo*, comparable to what we observed in chronically stressed mice. Systemic administration of mecamylamine was sufficient to virtually abolish this cellular maladaptation. For this last experiment, we could not deliver the drug via cannulas implanted

above the VTA as this prevented us from obtaining healthy brain sections for *ex vivo* recordings. This set of data has been added in Fig. 4 and prove that cholinergic inputs from the LDTg to the VTA drive cellular and behavioral adaptations to social stress via nicotinic receptors.

In addition, we have attempted a different strategy in order to inhibit LDTg projections in a target- and cell type-specific manner. For this, we acquired newly developed virus tools from McGovern Vector facility including retrograde herpes viruses expressing flipase in a Cre-dependent manner (HSV-hEF1a-LS1L-flpo) and flipase-dependent DREADDs (AAV8.2-hEF1a-fDIO-hM4D-mCherry). We injected the HSV in the VTA and the AAV in the LDTg of ChATCre mice. However, we observed that the resulting recombination is not specific to cholinergic neurons, and therefore not suitable to answer the reviewer concerns with this complementary approach.

Figure 1 Rebuttal: *ChATCre mice were injected with HSV-hEF1a-LS1L-flpo in the VTA and with AAV8.2-hEF1a-fDIO-hM4D-mCherry in the LDTg. Following three weeks of virus expression, immunofluorescence anti mCherry (red) and anti vAChT (green; vesicular acetylcholine transporter) was performed on LDTg slices. Arrow heads*

indicate cells that are mCherry positive and vAChT negative, indicating DREADD expression in non-cholinergic neurons.

2. Recent work has cast some doubt on the use of electrophysiological criteria to accurately classify VTA neurons as dopaminergic and non-dopaminergic. Broad action potentials have been used to identify DA neurons, but this criterion has been shown to exclude the majority of neurons identified as DA by other methods (see e.g. Cohen et al., 2012). Importantly, other work has shown that VTA DA neurons with different projection targets have different spike widths (Margolis et al., 2008). For example, DA neurons that project to the amygdala tend to be short. In brief: broad action potentials are likely DA, but this selection criterion excludes the majority of DA neurons and likely introduces bias. The authors should address these issues when describing their VTA recordings. Note: only a single sentence is given regarding the classification of DAergic recordings, citing a paper (Morel et al. 2017) that is not included in the references list.

We thank the reviewer for highlighting this issue and the missing reference. We have amended the method section and added a new figure (Supplementary Fig. 6) validating the recordings of DA neurons performed in this study. We also included additional references as suggested by the reviewer. (p. 17, §3)

3. While the authors provide evidence for the role of LDTg cholinergic neurons influencing VTA DA activity in states of stress, these results are not fully integrated into the more recent literature regarding the effect of chronic social defeat stress on dopaminergic firing. Of note, different stress models have demonstrated conflicting effects on DAergic activity, with chronic mild stress (CMS) exerting a decrease in DAergic activity, as opposed to the CSD-induced increase in activity. For example, see Chang & Grace (2014) Biol. Psychiatry and

supplementary figures in Tye et al. (2013) Nature. Conflicting results from these stress paradigms should be considered in the discussion section.

This is indeed an unresolved issue in the field and we have added a paragraph in our discussion (p. 12, §2) to encompass the literature on different stress paradigms of depression and the related changes in VTA DA neurons activity. We have amended the corresponding citations as suggested by the reviewer.

• **Minor points:**

1. Please identify the c-Fos+ neurons in Figure 1A via immunohistochemistry. Are they ChAT+ or vglut2+?

We conducted additional immunohistofluorescence experiments to monitor c-Fos expression after stress in cholinergic and glutamatergic LDTg neurons. We modified the original Fig. 1a to include these results that showed that acute social stress elicits a significant c-Fos expression in both cell types (see revised Fig. 1a).

2. Figure 1g: the authors don't give numbers for each condition independently. Please give numbers, and if there are fewer data points for the CNO animals please collect more data until there is equivalent statistical power as for the saline condition. For the CNO naive, in particular, there appear to be many fewer data points. Also, the interaction has technically not reached significance.

We have now detailed in each figure legend the number of cell and/or mice used in each experimental group condition. We also performed more recordings to equilibrate the numbers of neurons recorded in Fig. 1g as requested (see revised Fig. 1g).

3. Figure 4A: please give a time base in panel 3 (stressin/urocortin)

We have modified the figure to add time and voltage scale bars. Original Fig.4 is now revised Fig. 5.

4. The last panel of Figure 4A has cut off lines.

We have corrected this mistake in the revised Fig. 5A.

5. "Accordingly, we have shown that constitutive ablation of nAChRs is sufficient to prevent CSD-induced hyperactivity of DA neurons": This should include a citation.

We have added the reference Morel et al., 2017.

6. (p. 14): Morel et al., 2017 is not in the references list.

This reference has been added to the list.

Reviewer #2 (Remarks to the Author):

The study by Fernandez et al examines laterodorsal tegmentum (LDTg) input to VTA in stress-induced depressive like behavior. The authors use a multidisciplinary approach including DREADDs, in vivo and slice recordings, neuron subtype transgenic mice, and pharmacology combined with a behavioral model of stress-induced depression that has strong face validity for modeling depression. Overall the authors providing exciting data to demonstrate a role for cholinergic LDTg inputs in mediating the behavioral responses and characteristic VTA DA neuron firing, through CRF mediated mechanisms, in stress susceptible outcomes to chronic social defeat stress. This study would be of wide interest to the Neuroscience community especially those interested in circuit mechanism in depression.

While the results are compelling the conclusions could be strengthened further through additional experiments and analysis. See specific comments below.

1) Given recent studies demonstrating that CNO can be converted to clozapine it is important to include CNO controls in social defeat. Given that the authors observe no difference in behavior and DA neuron firing with CNO in the Vglut-hM4 group then this essentially acts as a control for CNO. It would be useful if the authors acknowledge this in the discussion.

This issue was also raised by Reviewer 3 (point 3). We performed additional experiments and showed that:

- acute administration of CNO does not modify firing and bursting activities of VTA DA neurons measured *in vivo* in anaesthetized mice.

- chronic administration of CNO during CSD does not prevent hyperactivity of VTA DA neurons measured *ex vivo*.

- chronic administration of CNO during CSD does not prevent the appearance of social aversion and anhedonia.

These results have been included in revised supplementary Fig. 3 and demonstrate the lack of effect of CNO alone in our behavioral and cellular readouts. We also included a paragraph in the discussion of our manuscript to discuss these key control experiments (p. 12, §3).

2) For the social interaction data in Figures 1d, 3b, 3e, and 4c the data should be analyzed by stress x treatment similar to the sucrose preference data. Please also show a between subjects comparison of saline vs. CNO (or CP376395) for the Target condition rather than comparing each group to its no Target score.

We thank the reviewer for his/her suggestion. We have performed additional statistical analyses addressing stress x treatment interactions. We report these values in the figure legends. We have also performed between subjects comparisons for the target condition, which are now depicted in the revised figures.

3) In Figure 2E the authors demonstrate that LDTg-VTA Vglut2+ neurons display increased firing in CSDS mice. However, their DREADD experiments in Figure 3E and F target all Vglut2+ LDTg neurons. A more specific targeting of LDTg-VTA Vglut2+ neurons (i.e. CNO application to LDTg Vglut2+ terminals in VTA) could provide a more definitive conclusion that these neurons do not mediate the behavioral and DA neuron firing effects in CSDS mice.

We thank the reviewer for his/her comments. As described in the response to Reviewer 1 major point 1, we demonstrated that chemogenetic-mediated activation of LDTg inputs to the VTA combined with a subthreshold stress is sufficient to promote cellular and behavioral maladaptations. Although, this strategy allows activation of both cholinergic and glutamatergic neurons that project to the VTA, blocking cholinergic transmission via nicotinic receptors was sufficient to fully prevent this effect. Therefore, we concluded that acetylcholine release is key to gate stress impact onto VTA DA neurons. This is consistent with our original submitted data showing that inhibition of cholinergic LDTg neurons markedly reduces CSD-induced increases in AMPA-R/NMDA-R ratio (supplementary Fig. 5c). This implies that acetylcholine release is required to shape glutamate signals. In light of these accumulated evidence and the fact that the VTA receives different sources of glutamate inputs, we believe that inhibiting LDTg glutamatergic terminals in the VTA will not produce stress relief. Nevertheless, we do agree with the reviewer that, even though the mechanisms are at present unclear, the glutamatergic synapse undergoes clear remodeling that is likely to impact DA neurons activity. This point has been made clear in the results section (p. 7, §1) and in the discussion (p. 11, §2) to avoid misleading of the reader and oversimplification of the message conveyed by our study. We do provide strong evidence for bidirectional

implication of cholinergic mechanisms but cannot rule out glutamate contribution to these processes.

4) Should the authors identify an effect with direct DREADD inhibition of LDTg-VTA Vglut2+ neurons or terminals (see above comment) it would be useful to examine the response of this population to Stressin and Urocortin as was performed in ChAT neurons in Figure 4.

We thank the reviewer for this suggestion. We have performed additional recordings that have been included in revised Figure 5 and have also modified the text (p. 9, §1). These results clearly show that neither stressin I nor urocortin III activate LDTg glutamatergic neurons. The mechanism by which these glutamatergic neurons increase firing after CSD remains therefore elusive. As suggested by Reviewer 3 point 4, we have now included a schematic to summarize our findings and the unresolved questions (Fig. 6).

5) To further determine if the restoration of social interaction and sucrose preference, using a CRF-R1 antagonist into LDTg in Figure 4c and 4d, is mediated through the ChAT LDTg-VTA neurons it would be useful to examine neuron excitability in this population as in Figure 2e. Similar analysis in LDTg-VTA Vglut2+ neurons will be important if DREADD inhibition of the projection neuron subtype alters behavior (see comment #3).

This is an interesting point that we have previously attempted to address. We did heavily attempt to record LDTg neurons following long-term local infusion. This failed due to poor quality of brain slices obtained, perhaps related to the detachment of the cannulas before slicing. Even in the few cases where we managed to record cells, we corroborated that cells were unhealthy therefore results not reliable.

However, we conducted new experiments to assess the excitability of LDTg cholinergic neurons in ChATCre mice following repeated DREADD-mediated inhibition. We show that cells expressing hM4, from mice that received CNO but not saline, did not increase their firing frequencies following CSD. These results have been included in revised supplementary Fig. 5d.

6) Minor comment- “neurons hyperexcitability” on page 7 line 3 and page 8 line 13 should be “neuron.”

We have now corrected the text.

Reviewer #3 (Remarks to the Author):

This manuscript discusses an interesting circuit that may mediate depression phenotypes, especially social avoidance induced by chronic social defeat. It does a nice job of dissecting the circuitry of LDTg inputs to VTA DA neurons in the context of social defeat, however questions remain about how to place these findings within the context of a broader, somewhat controversial literature. More data on the mechanism of how acetylcholine released by the LDTg influences the DA circuit could also increase the impact of the findings, although I don't think it should necessarily be required for publication.

• Major Concerns:

1. How much of this data is specific to the social defeat stress paradigm? The manuscript references work by Chaudhury et al showing that social defeat stress specifically can cause

increases in VTA DA neuron firing (in VTA neurons projecting to NAc only). However, there is an abundance of work suggesting that VTA DA neuron firing promotes motivational behavior and indeed a paper published side-by-side with Chaudhury et al – Tye et al. – is not referenced or discussed. It is hard to place this manuscript's findings in the broader context of depression without understanding why the social defeat paradigm seems to differ from other stress/depression paradigms, such as chronic mild stress, in terms of its effects on VTA DA excitability. Additionally, Lammel et al (Nature, 2012) have shown that stimulation of LDTg inputs to VTA induces place preference, so why would inhibition reverse depression symptoms? The authors must discuss these issues and limit their conclusions and interpretation accordingly.

We thank the reviewer for his/her comments. Reviewer 1 also raised the point of different rodent models of depression that differently impact VTA DA neurons. This is indeed an unresolved issue in the field and we have added a paragraph in our discussion (p. 12, §2) to encompass the literature on different stress paradigms of depression and the related changes in VTA DA neurons activity. We have amended the corresponding citations as suggested by the reviewer.

For the second point, previous reports have shown that optogenetic activation of LDTg terminals in the VTA (Lammel et al., 2012) or direct phasic stimulation of VTA DA neurons (Tsai et al., 2009) elicit conditioned place preference. This indicates that this pathway can at least support rewarding processes. Nevertheless, a wealth of literature indicates that a variety of aversive stimuli can alter VTA DA neurons firing (Barik et al., 2013; Brischoux et al., 2009; Cao et al., 2010; Eddine et al., 2015; Tye et al., 2013; Valenti et al., 2011), and our current study also clearly shows that social stress markedly impacts LDTg neurons projecting to the VTA. This suggests that, in the context of stressful events, LDTg and VTA DA neurons can convey salient information and not only reward. We have modified our discussion to highlight this point (p. 12, §1)

2. What are the expression patterns of CRF1 and CRF2 receptors in LDTg?

There is scarce evidence regarding the patterns of expression of CRF1 and CRF2 receptors in the LDTg. Two studies used immunohistochemical approaches and detected the presence of CRF-R1 in cholinergic neurons (Crawley et al., 1985; Sauvage and Steckler, 2001). As described in response to Reviewer 2 point 4, we carried out additional *ex vivo* recordings of glutamatergic LDTg neurons and show that they do not respond to selective CRF-R1 or CRF-R2 agonists. Therefore, the functional readouts we provide in both cholinergic and glutamatergic LDTg neurons, in addition to the existing literature, indicate that CRF release during the stress response will impact LDTg function via CRF-R1 located on cholinergic neurons. We now provide a schematic (Fig. 6) to summarize our findings.

3. Given recent findings that clozapine is in fact the active, BBB-crossing metabolite of CNO that mediates chemogenetic behavioral effects, and that clozapine acts as a dopamine antagonist, I would like to see more data on the effects of CNO injection alone on in vivo VTA firing patterns and behavior in CSD vs naïve mice. The authors do the proper control of injecting CNO in naïve mice in behavioral experiments, however, I don't see injections of CNO in non-hM4D-expressing mice, which would aid in the interpretation of their results. This could be especially key for the interpretation of sucrose preference in Fig 1e, which does not look like nearly as robust an effect as for social interaction time, and perhaps for interpreting the lack of effect when manipulating LDTg glutamatergic neurons.

This issue was also raised by Reviewer 2 point 1. As detailed, above we have performed additional experiments both *in vivo* and *ex vivo* and showed the innocuity of CNO in the behavioral and cellular readouts used in this study. This is discussed in p. 12, §3.

4. The connection between VTA DA hyperexcitability and cholinergic inputs from LDTg remains a bit confusing. The authors do acknowledge in the conclusion that more work needs to be done to establish the site of glutamatergic synaptic plasticity onto DA neurons and the relationship between synaptic and intrinsic plasticity, which I think is understandable. A summary schematic could be helpful to illustrate both the conclusions and areas for further study.

We thank the reviewer for his/her comments and added a schematic to summarize our results and highlight the points that will require further investigations. See new Fig. 6.

5. The authors state “We have shown that constitutive ablation of nAChRs is sufficient to prevent CSD-induced hyperactivity of DA neurons,” but is there a reference for this statement? Was the ablation of nAChRs expressed by DA neurons specifically?

We have now added the missing reference (Morel et al., 2017). In this previous study, we used constitutive double knock-out mice for both $\alpha 7$ and $\beta 2$ nicotinic acetylcholine receptor (nAChRs) subunits to globally address the role of nAChRs in stress-induced adaptations. This was therefore not restricted to DA neurons. The present study clearly implicates LDTg cholinergic inputs to the VTA that signal through VTA nAChRs to trigger cellular and behavioural maladaptations.

References

Barik, J., Marti, F., Morel, C., Fernandez, S.P., Lanteri, C., Godeheu, G., Tassin, J.-P., Mombereau, C., Faure, P., and Tronche, F. (2013). Chronic Stress Triggers Social Aversion via Glucocorticoid Receptor in Dopaminergic Neurons. *Science* 339, 332–335.

Brischoux, F., Chakraborty, S., Brierley, D.I., and Ungless, M.A. (2009). Phasic excitation of dopamine neurons in ventral VTA by noxious stimuli. *Proc. Natl. Acad. Sci. U. S. A.* 106, 4894–4899.

Cao, J.-L., Covington, H.E., Friedman, A.K., Wilkinson, M.B., Walsh, J.J., Cooper, D.C., Nestler, E.J., and Han, M.-H. (2010). Mesolimbic Dopamine Neurons in the Brain Reward Circuit Mediate Susceptibility to Social Defeat and Antidepressant Action. *J. Neurosci.* 30, 16453–16458.

Crawley, J.N., Olschowka, J.A., Diz, D.I., and Jacobowitz, D.M. (1985). Behavioral investigation of the coexistence of substance P, corticotropin releasing factor, and acetylcholinesterase in lateral dorsal tegmental neurons projecting to the medial frontal cortex of the rat. *Peptides* 6, 891–901.

Eddine, R., Valverde, S., Tolu, S., Dautan, D., Hay, A., Morel, C., Cui, Y., Lambollez, B., Venance, L., Marti, F., et al. (2015). A concurrent excitation and inhibition of dopaminergic subpopulations in response to nicotine. *Sci. Rep.* 5, 8184.

Lammel, S., Lim, B.K., Ran, C., Huang, K.W., Betley, M.J., Tye, K.M., Deisseroth, K., and Malenka, R.C. (2012). Input-specific control of reward and aversion in the ventral tegmental area. *Nature* 491, 212–217.

Morel, C., Fernandez, S.P., Pantouli, F., Meye, F.J., Marti, F., Tolu, S., Parnaudeau, S., Marie, H., Tronche, F., Maskos, U., et al. (2017). Nicotinic receptors mediate stress-nicotine detrimental interplay via dopamine cells' activity. *Mol. Psychiatry*.

Sauvage, M., and Steckler, T. (2001). Detection of corticotropin-releasing hormone receptor 1 immunoreactivity in cholinergic, dopaminergic and noradrenergic neurons of the murine basal forebrain and brainstem nuclei – potential implication for arousal and attention. *Neuroscience* 104, 643–652.

Tsai, H.-C., Zhang, F., Adamantidis, A., Stuber, G.D., Bonci, A., Lecea, L. de, and Deisseroth, K. (2009). Phasic Firing in Dopaminergic Neurons Is Sufficient for Behavioral Conditioning. *Science* 324, 1080–1084.

Tye, K.M., Mirzabekov, J.J., Warden, M.R., Ferenczi, E.A., Tsai, H.-C., Finkelstein, J., Kim, S.-Y., Adhikari, A., Thompson, K.R., Andalman, A.S., et al. (2013). Dopamine neurons modulate neural encoding and expression of depression-related behaviour. *Nature* 493, 537–541.

Valenti, O., Lodge, D.J., and Grace, A.A. (2011). Aversive Stimuli Alter Ventral Tegmental Area Dopamine Neuron Activity via a Common Action in the Ventral Hippocampus. *J. Neurosci.* 31, 4280–4289.

Reviewers' comments:

Reviewer #1 (Remarks to the Author):

I thank the authors for addressing many of my concerns. Unfortunately, the major concern voiced in my previous review – evidence for specific involvement of the cholinergic LDTg-VTA projection in mediating stress-induced behaviors, referenced in the title of the paper – has not yet been convincingly addressed.

It is very much appreciated that the authors attempted to specifically target the cholinergic LDTg-VTA projection with a flp/cre strategy. Unfortunately, this method did not result in expression specific to LDTg-VTA cholinergic neurons. Therefore, they instead used a CAV approach to transduce the LDTg-VTA projection, which has both cholinergic and glutamatergic neurons. Exciting this projection in a subthreshold social defeat stress model potentiated the impact of stress on behavior and VTA physiology. This effect that was blocked by nicotinic receptor antagonism in the VTA, showing that this effect depends on acetylcholine.

Although this data is suggestive, it is not proof that the cholinergic LDTg-VTA projection is responsible. It is possible that LDTg-VTA glutamate projection neurons were responsible, and that the behavioral and VTA physiology consequences of stimulating this glutamatergic input require local VTA cholinergic tone. This is not unlikely, since there are presynaptic nicotinic acetylcholine receptors on glutamatergic axons in the VTA. There are other major cholinergic inputs to the VTA, including the input from the pedunculopontine tegmental nucleus. Also, the authors did not perform immunostaining on the LDTg cell bodies in this experiment, so it is unknown whether this CAV approach selectively targets cholinergic or glutamatergic neurons or both. This is an issue of some concern, as CAV vectors appear to have some tropism.

Major request: In order for the title and wording in the paper to stand, the authors need to directly show that the specific cholinergic LDTg-VTA projection mediates stress-induced behaviors. Although the flp/cre strategy did not work, the authors could achieve the same objective by transducing cholinergic neurons in the LDTg with DREADDS (ChAT-cre mouse, cre-dependent DREADDS), and then locally infuse CNO into the VTA to specifically inhibit or activate this projection (similar to reviewer 2's point 3). There have been several recent studies utilizing terminal CNO application for projection-specific manipulations (e.g. Burnett and Krashes J Neuro 2016). A behavioral result would be sufficient if this manipulation makes VTA physiology impossible.

The authors have presented a wealth of evidence that LDTg ChAT neurons play an essential role in mediating behavioral and VTA physiology responses to social defeat stress, this point is exceptionally well supported. The only remaining issue is the specific role of the cholinergic LDTg-VTA projection.

Reviewer #2 (Remarks to the Author):

The authors have addressed all concerns.

Reviewer #3 (Remarks to the Author):

The authors have addressed my questions and concerns and I believe these results will be of interest to the general community, especially those interested in stress-related neural adaptations. I recommend publication.

Reviewer #1 (Remarks to the Author):

I thank the authors for addressing many of my concerns. Unfortunately, the major concern voiced in my previous review – evidence for specific involvement of the cholinergic LDTg-VTA projection in mediating stress-induced behaviors, referenced in the title of the paper – has not yet been convincingly addressed.

It is very much appreciated that the authors attempted to specifically target the cholinergic LDTg-VTA projection with a flp/cre strategy. Unfortunately, this method did not result in expression specific to LDTg-VTA cholinergic neurons. Therefore, they instead used a CAV approach to transduce the LDTg-VTA projection, which has both cholinergic and glutamatergic neurons. Exciting this projection in a subthreshold social defeat stress model potentiated the impact of stress on behavior and VTA physiology. This effect that was blocked by nicotinic receptor antagonism in the VTA, showing that this effect depends on acetylcholine.

Although this data is suggestive, it is not proof that the cholinergic LDTg-VTA projection is responsible. It is possible that LDTg-VTA glutamate projection neurons were responsible, and that the behavioral and VTA physiology consequences of stimulating this glutamatergic input require local VTA cholinergic tone. This is not unlikely, since there are presynaptic nicotinic acetylcholine receptors on glutamatergic axons in the VTA. There are other major cholinergic inputs to the VTA, including the input from the pedunculopontine tegmental nucleus. Also, the authors did not perform immunostaining on the LDTg cell bodies in this experiment, so it is unknown whether this CAV approach selectively targets cholinergic or glutamatergic neurons or both. This is an issue of some concern, as CAV vectors appear to have some tropism.

Major request: In order for the title and wording in the paper to stand, the authors need to directly show that the specific cholinergic LDTg-VTA projection mediates stress-induced behaviors. Although the flp/cre strategy did not work, the authors could achieve the same objective by transducing cholinergic neurons in the LDTg with DREADDS (ChAT-cre mouse, cre-dependent DREADDS), and then locally infuse CNO into the VTA to specifically inhibit or activate this projection (similar to reviewer 2's point 3). There have been several recent studies utilizing terminal CNO application for projection-specific manipulations (e.g. Burnett and Krashes *J Neuro* 2016). A behavioral result would be sufficient if this manipulation makes VTA physiology impossible.

The authors have presented a wealth of evidence that LDTg ChAT neurons play an essential role in mediating behavioral and VTA physiology responses to social defeat stress, this point is exceptionally well supported. The only remaining issue is the specific role of the cholinergic LDTg-VTA projection.

We thank the referee for her/his comments and performed a new experiment to address this issue. As suggested by the reviewer, we injected ChATCre mice with a AAV-hSyn-DIO-hM3D-mcherry in the LDTg and implanted guides above the VTA for local CNO delivery. This allows us to combine a chemogenetic activation of LDTg cholinergic terminals within the VTA with a subthreshold social defeat (SubSD) paradigm. This double-hit strategy resulted in the appearance of social aversion 24h after in mice receiving CNO but not vehicle. This result has been added to revised Figure 4d. We also modified the results (p. 8, §1) and amended the method section (p. 18, §1).

Overall, this new data demonstrates that LDTg cholinergic projection to the VTA is key for maladaptations induced by chronic social stress.

Reviewer #2 (Remarks to the Author):

The authors have addressed all concerns.

We thank the referee for her/his helpful feedback.

Reviewer #3 (Remarks to the Author):

The authors have addressed my questions and concerns and I believe these results will be of interest to the general community, especially those interested in stress-related neural adaptations. I recommend publication.

We thank the referee for her/his helpful feedback.